# A Dual-View Contrastive Learning Framework for Heterogeneous Graph Representation Learning

## Abstract

Heterogeneous graph representation learning leverages the rich semantics and complex structural relationships within heterogeneous graphs. However, existing methods often fail to capture long-range semantic dependencies and localized structural patterns simultaneously. Therefore, we propose a novel Dual-View Contrastive Learning framework (DVCL) for heterogeneous graph representation learning. Specifically, the Graph Schema View Module (GSVM) is conducted to model the structural dependencies by leveraging a relational graph neural network with type-aware message passing and adaptive residual connections. Then, the Semantic Meta-Path Mamba Module (SMPMM) is designed to capture high-order semantic dependencies through a globally enhanced Mamba backbone, equipped with multi-resolution fusion and directional positional encodings. Moreover, a dynamic bidirectional contrastive learning is constructed to integrate the semantic view and structural view, treating each view as a learnable augmentation of the other to ensure robust and complementary representations. Extensive experiments on four datasets demonstrate that the proposed method consistently outperforms state-of-the-art methods, in terms of classification and clustering tasks.

## 1 Introduction

Heterogeneous graphs, which consist of multiple types of nodes and edges, have emerged as fundamental abstractions for representing complex and richly structured data across various domains, including social networks Singh & Verma (2025), bibliographic databases Yu et al. (2024), and recommendation systems Khan et al. (2025). By explicitly modeling heterogeneity in both node and relation types, heterogeneous graph-based methods are capable of capturing nuanced semantics and contextual relationships. Effectively learning expressive and generalizable node representations from such data is crucial for enabling a broad range of downstream tasks, such as node classification Li et al. (2024b) and clustering Wang et al. (2024).

Recent heterogeneous graph representation learning methods can be grouped into three main paradigms: supervised learning, generative learning, and contrastive learning. Supervised learning-based methods typically rely on labeled data and type-aware neural architectures to perform end-to-end training for targeted tasks Shi et al. (2023). These methods leverage relation-specific message passing and hierarchical aggregation schemes to capture local structures and semantic dependencies. Although capable of high performance with ample labeled data, these models are prone to overfitting on specific tasks, which hinders their application in low-label or inductive settings Li et al. (2023b); Liao et al. (2022). In order to deal with limited supervision, generative learning-based methods aim to reconstruct graph structures or feature distributions to learn node representations Gupta & Bedathur (2022). While they can generalize better with limited labels, these methods often suffer from training instability and scalability issues Xia et al. (2025); Gupta & Bedathur (2022).

Supervised methods often suffer from strong label dependency, while generative approaches are prone to instability due to reconstruction-based objectives. Therefore, contrastive learning has emerged as a powerful self-supervised paradigm for heterogeneous graph representation learning Chen et al. (2023); Moradi et al. (2025); Dang et al. (2025); Wang et al. (2025b). By maximizing agreement between augmented views of a graph without requiring explicit labels, contrastive meth-

ods effectively mitigate the overfitting issues of supervised models and the training instability of generative models Chen et al. (2023). For example, HeCo Wang et al. (2021) introduces a co-contrastive framework that aligns representations from meta-path and schema views, and HeCo++ Liu et al. (2023) proposes hierarchical contrastive objectives to disentangle multi-level relational patterns.

Despite the remarkable progress of contrastive learning on heterogeneous graphs, two fundamental challenges remain unresolved. (1) Existing contrastive frameworks often treat the heterogeneous graph as a whole and apply augmentation or multi-view fusion at the global level Li et al. (2023a). Some methods first perform multi-graph fusion followed by contrastive learning, which overlooks the uniqueness of each heterogeneous schema and its contrastive representation Ke et al. (2023). Others conduct result-level fusion for downstream tasks, but ignore the shared information and complementary patterns across different heterogeneous substructures Zheng et al. (2024); Chen et al. (2023). Consequently, multi-perspective heterogeneous representations are ignored and principled mechanism is absent for searching relation-specific views and exploring schema-level semantics. (2) Existing methods struggle to capture high-order semantic dependencies. GCNs mainly aggregate local neighborhoods and fail on long-range semantics Qian & Yin (2024); Liang et al. (2023), while Transformer-based models extend the receptive field but incur prohibitive quadratic complexity on large heterogeneous graphs Li et al. (2022); Sharma et al. (2025). As a result, multi-hop semantic relations (e.g., author–paper–venue) are often underexplored, limiting the expressive power of current approaches Wan & Ding (2024); Wang et al. (2023b).

To address these limitations, we propose a novel Dual-View Contrastive Learning (DVCL) framework that bridges the gap between local structures and long-range dependencies by dynamically integrating two complementary perspectives of the graph. Specifically, the Graph Schema View Module (GSVM) leverages a type-aware Heterogeneous Relational Graph Convolutional Network (HeteroRGCN) to capture fine-grained local structural dependencies and schema-level information. In parallel, the Semantic Meta-Path Mamba Module (SMPMM) is designed to model high-order semantic dependencies along meta-paths, incorporating directional positional encodings and multi-resolution fusion to effectively capture long-range relationships. Finally, a bidirectional contrastive loss aligns the two views, maximizing their agreement while preserving their distinct inductive biases. This dual-view design enables the learning of robust and transferable node embeddings, facilitating diverse downstream applications in heterogeneous graph scenarios.

The main contributions of this paper are summarized as follows: (1) We propose a dual-view contrastive learning framework that encodes local relational structures via a structure-aware relational model and then captures long-range semantic dependencies through a Mamba-based semantic model, enabling comprehensive representation learning for heterogeneous graphs. (2) We present a heterogeneous graph framework with three key components: a GSVM for relational schema modeling via type-aware message passing , SMPMM for capturing meta-path semantics , and a bidirectional contrastive module for robust representation learning. (3) Through extensive experiments and ablation studies, our approach achieves state-of-the-art performance in four datasets in terms of classification and clustering task.

## 2 APPROACH

In this section, we introduce the framework of Dual-View Contrastive Learning (DVCL) framework, which consists of three core components: the Graph Schema View Module (GSVM), the Semantic Meta-Path Mamba Module (SMPMM), and the Bidirectional Contrastive Learning Module. The details can be found in Figure 1.

### 2.1 DATA PRE-PROCESSING

In heterogeneous graphs, node features across different types often differ in both dimensionality and semantics, which hinders direct integration. Therefore, we design a type-aware projection that maps raw features of each node type into a unified latent space.

Formally, let $G = (V, E, \mathcal{A}, R)$ denote a heterogeneous graph, where $V$ is the set of nodes, $E \subseteq V \times V$ is the set of edges, $\mathcal{A}$ is the set of node types (e.g., author, paper, venue), and $R$ is the set of relation types. For each node type $a \in \mathcal{A}$, let $\mathbf{X}^{(a)} \in \mathbb{R}^{N_a \times h}$ denote the input feature matrix, where $N_a$ is the number of nodes of type $a$, and $h$ is the input feature dimension.

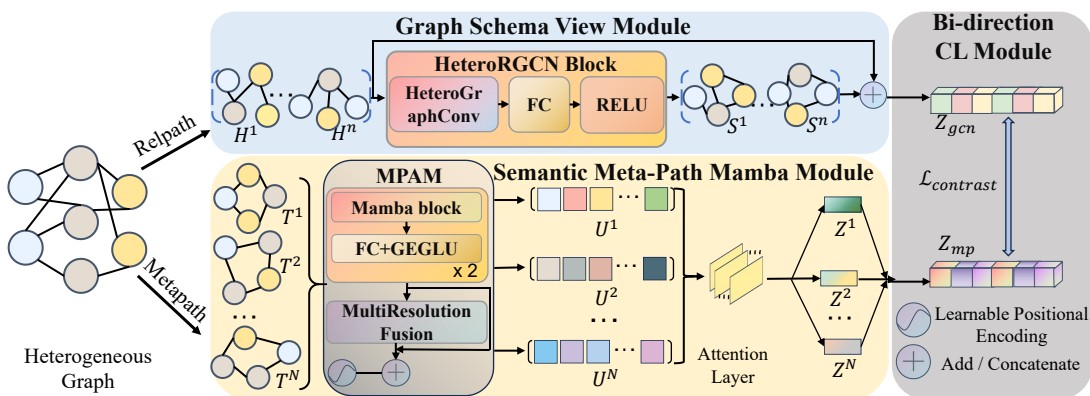

Figure 1: The DVCL framework consists of three components: (1) **Graph Schema View Module (GSVM)** employs a Relational Graph Convolutional Network (R-GCN) to capture type-aware structural dependencies, producing a topology-aware structure view. (2) **Semantic Meta-Path Mamba Module (SMPMM)** extracts high-order semantic dependencies from meta-path-based sequences to conduct sequence view. (3) **Bidirectional Contrastive Learning Module** aligns the structure and sequence view, using a symmetric contrastive loss to enforce cross-view consistency while maintaining their distinctiveness for robust representation learning.

---

**Algorithm 1** The procedure of the proposed DVCL framework.

---

**Input**: Heterogeneous graph $G = (V, E, \mathcal{A}, R)$, raw features $\{\mathbf{X}^{(a)}\}_{a \in \mathcal{A}}$, meta-paths $\{1, \ldots, P\}$;
**Output**: Final node embeddings $\mathbf{Z} \in \mathbb{R}^{|V| \times h}$;

1: Obtain type-aware features $\widetilde{\mathbf{H}}^{(a)}$ by type-specific projection and activation by Eq. 2;
2: Compute structural embeddings $\mathbf{Z}_{\text{gcn}}$ via R-GCN on $\widetilde{\mathbf{H}}$ by Eq. 6;
3: **for** each meta-path $p$ **do**
4:     Compute semantic embeddings $\mathbf{Z}^{(p)}$ using SMPMM on sequences from $\widetilde{\mathbf{H}}$ by Eq. 8;
5: **end for**
6: Fuse semantic embeddings $\mathbf{Z}_{\text{mp}} = \text{Fuse}(\{\mathbf{Z}^{(p)}\})$ by Eq. 9;
7: Project $\mathbf{Z}_{\text{gcn}}, \mathbf{Z}_{\text{mp}}$ into contrastive space and compute bidirectional contrastive loss $\mathcal{L}_{\text{contrast}}$ by Eq. 14;
8: **return** final node embeddings $\mathbf{Z}$.

---

To unify these heterogeneous features, we apply a type-specific linear projection:

$$\mathbf{H}^{(a)} = \mathbf{X}^{(a)} \mathbf{W}^{(a)} + \mathbf{B}^{(a)}, \tag{1}$$

where $\mathbf{W}^{(a)} \in \mathbb{R}^{h \times d}$ is a learnable projection matrix, and $\mathbf{B}^{(a)} \in \mathbb{R}^{N_a \times d}$ denotes the learnable bias matrix. All projection matrices are initialized using Xavier normal initialization Glorot & Bengio (2010) with a gain of $\sqrt{2}$.

To enhance expressiveness and mitigate overfitting, we apply an ELU activation and shared dropout:

$$\widetilde{\mathbf{H}}^{(a)} = \text{Dropout}(\text{ELU}(\mathbf{H}^{(a)})), \quad \forall a \in \mathcal{A}. \tag{2}$$

Here, $\mathbf{X}^{(a)}$ represents raw features of type-$a$ nodes, and $\widetilde{\mathbf{H}}^{(a)} \in \mathbb{R}^{N_a \times d}$ are the resulting type-aware representations, projected into a shared latent space. This transformation improves feature alignment, reduces noise and redundancy, and serves as the input to two key modules: the Graph Schema View Module (GSVM), which captures structural dependencies via type-aware message passing, and the Semantic Meta-Path Mamba Module (SMPMM), which models high-order semantic correlations along meta-paths. Together, these modules enable the DVCL framework to jointly exploit structural and semantic cues across diverse node types.

## 2.2 GRAPH SCHEMA VIEW MODULE

Most current contrastive approaches treat heterogeneous graphs in an overly unified manner, either by merging multiple graph instances prior to training or by aggregating outputs after the contrastive stage Li et al. (2023a). These coarse-level designs dilute the unique contributions of individual relations and schemas, while also neglecting the complementary cues embedded within local substructures Ke et al. (2023). Consequently, the rich diversity of heterogeneous representations remains insufficiently exploited, and an effective framework for relation-sensitive view generation and schema-aware semantic modeling has yet to be established. To overcome this gap, we propose the **Graph Schema View Module (GSVM)**, which explicitly captures the multi-relational topology of heterogeneous graphs. Concretely, GSVM enriches meta-path semantics with detailed structural signals, yielding more informative and expressive node embeddings.

Specifically, GSVM is built upon R-GCN to propagate and aggregate information across different relation types. For a node $u$ at layer $l$, the update rule is:

$$\mathbf{H}_u^{(l)} = \sigma \left( \sum_{r \in R} \sum_{v \in \mathcal{N}_r(u)} \frac{1}{c_{uv}^{(r)}} \mathbf{W}_r^{(l)} \mathbf{H}_v^{(l-1)} + \mathbf{W}_u^{(l)} \mathbf{H}_u^{(l-1)} \right), \tag{3}$$

where $\mathcal{N}_r(u)$ denotes the neighbors of $u$ under relation $r$, $\mathbf{W}_r^{(l)}$ and $\mathbf{W}_u^{(l)}$ are learnable weights, $c_{uv}^{(r)}$ is a normalization factor, and $\sigma$ is a nonlinear activation. The initial embeddings $\mathbf{H}_u^{(0)}$ are drawn from the type-specific projected features $\widetilde{\mathbf{H}}^{(a)}$ according to $u$'s type $a \in \mathcal{A}$. All initial embeddings form the matrix $\mathbf{S}^{(0)} \in \mathbb{R}^{|V| \times h}$, with rows $\mathbf{S}^{(0)}[u, :] = \mathbf{H}_u^{(0)}$.

Stacking $L$ R-GCN layers yields

$$\mathbf{S}^{(l)} = \mathrm{ELU}\big(\mathrm{RGCNLayer}_l(\mathbf{S}^{(l-1)})\big), \quad l = 1, \ldots, L, \tag{4}$$

optionally followed by normalization and dropout for stability.

We adopt dual residual connections:

$$\mathbf{S}^{(l)} \leftarrow \mathbf{S}^{(l)} + \mathbf{S}^{(l-1)}, \quad \mathbf{S}^{(\mathrm{out})} \leftarrow \mathbf{S}^{(L)} + \mathbf{S}^{(0)}. \tag{5}$$

The final structural embeddings are then obtained as:

$$\mathbf{Z}_{\mathrm{gcn}} = \mathbf{S}^{(\mathrm{out})} \in \mathbb{R}^{|V| \times h}, \tag{6}$$

These embeddings effectively encode the relational schema of the graph, which are later aligned with semantic representations to enable joint learning of both structural and semantic information.

## 2.3 SEMANTIC META-PATH MAMBA MODULE

Accurately modeling intricate semantic structures in heterogeneous graphs remains a challenging task. Conventional message-passing mechanisms are restricted to local neighborhoods, making it difficult to aggregate signals across multiple hops Qian & Yin (2024); Liang et al. (2023). In contrast, approaches designed to capture long-distance interactions often encounter severe scalability issues when applied to large-scale graphs Li et al. (2022); Sharma et al. (2025). Consequently, many crucial multi-hop dependencies—such as the relations among authors, papers, and publication venues—are insufficiently utilized, limiting the expressive power of current models Wan & Ding (2024); Wang et al. (2023b). Specifically, (i) their computational demands escalate with the length of meta-paths Fu et al. (2020); Li et al. (2024a); Zhong et al. (2022), and (ii) attention mechanisms frequently mix heterogeneous signals, obscuring distinct semantic patterns Wang et al. (2021; 2023b). While contrastive paradigms attempt to mitigate this issue by enforcing consistency between multiple views, they predominantly operate at the view level and neglect fine-grained semantic separation. In summary, existing methodologies are either inefficient for modeling long-range dependencies or ineffective at maintaining semantic diversity, underscoring the need for a more robust solution.

Therefore, we introduce the **Semantic Meta-Path Mamba Module (SMPMM)**, which leverages the Mamba architecture, a recent state-space model with linear-time complexity and strong sequence modeling capabilities. This design effectively avoids the quadratic complexity of attention while capturing rich, compositional semantics along meta-paths.

Let $\mathcal{M} = \{1, 2, \ldots, P\}$ be a set of predefined meta-paths. For each meta-path $p \in \mathcal{M}$, we construct input sequences for a subset of target nodes $V_p \subseteq V$ by retrieving node features from the pre-processed type-aware representations $\{\widetilde{\mathbf{H}}^{(a)}\}_{a \in \mathcal{A}}$. This process yields an input tensor $\mathbf{T}^{(p)} \in \mathbb{R}^{|V_p| \times l \times h}$, where $|V_p|$ is the number of target nodes, $l$ is the length of meta-path $p$, and $h$ is the shared hidden dimension. Each sequence is passed through a stack of $B$ Mamba blocks specific to path $p$. To enhance semantic expressiveness, we apply a Gated GELU Feedforward (GEGLU) layer, which uses a gating mechanism to adaptively modulate semantic signals.

To capture both local and global patterns, a multi-resolution fusion block is inserted after every two Mamba layers. This block combines depthwise separable convolutions for local context and global average pooling for long-range semantics. We also incorporate learnable positional encodings $\{\mathbf{P}_j \in \mathbb{R}^d\}_{j=1}^{l}$ to preserve the directional semantics along the meta-path:

$$\mathbf{U}_{:,j,:}^{(p)} \leftarrow \mathbf{U}_{:,j,:}^{(p)} + \mathbf{P}_j. \tag{7}$$

The output sequence $\mathbf{U}^{(M_p)}$ is aggregated into a fixed-size vector per node and then projected into the unified latent space to get the semantic embedding for meta-path $p$:

$$\mathbf{Z}^{(p)} = \mathrm{Proj}_p(\mathrm{Aggregate}(\mathbf{U}^{(p)})) \in \mathbb{R}^{|V_p| \times h}. \tag{8}$$

Finally, the outputs of all meta-path encoders are fused via an attention pooling or a learnable weighted sum to obtain the final semantic embedding, $\mathbf{Z}_{\mathrm{mp}}$:

$$\mathbf{Z}_{\mathrm{mp}} = \mathrm{Fuse}(\{\mathbf{Z}^{(p)}\}_{p=1}^{P}) \in \mathbb{R}^{|V| \times h}. \tag{9}$$

This semantic view, $\mathbf{Z}_{\mathrm{mp}}$, complements the structural view, $\mathbf{Z}_{\mathrm{gcn}}$, and together they form the basis for our contrastive learning objective.

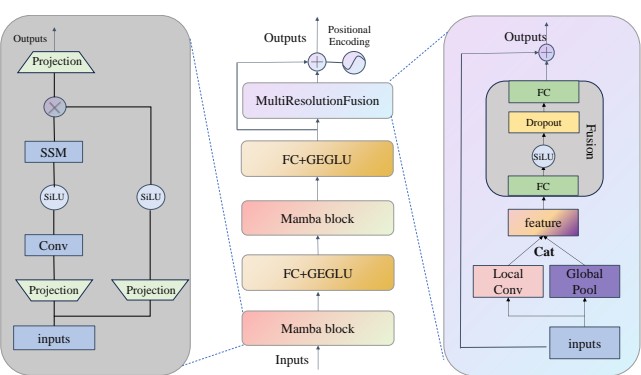

Figure 2: Architecture of the Meta-Path–Aware Mamba encoder (MPAM).

## 2.4 BIDIRECTIONAL CONTRASTIVE LEARNING

To effectively integrate the complementary structural and semantic information, we propose a novel bidirectional contrastive learning framework. Instead of a simple concatenation or fusion, we treat the semantic embeddings $\mathbf{Z}_{\mathrm{mp}}$ and the structural embeddings $\mathbf{Z}_{\mathrm{gcn}}$ as two distinct yet complementary views of the same underlying graph. Our goal is to enforce mutual alignment between these views for each node.

Before computing similarity, each view is passed through a separate two-layer projection head, which maps the embeddings to a shared contrastive space:

$$\tilde{\mathbf{Z}}_{\mathrm{mp}} = \phi_{\mathrm{mp}}(\mathbf{Z}_{\mathrm{mp}}), \quad \tilde{\mathbf{Z}}_{\mathrm{gcn}} = \phi_{\mathrm{gcn}}(\mathbf{Z}_{\mathrm{gcn}}), \tag{10}$$

where $\phi_{\mathrm{mp}}$ and $\phi_{\mathrm{gcn}}$ are feedforward networks with ELU activations. We then compute the scaled dot-product similarity between all pairs of projected embeddings:

$$E_{ij}^{\mathrm{mp} \to \mathrm{gcn}} = \frac{\tilde{\mathbf{Z}}_{\mathrm{mp}}^{(i)} \cdot \tilde{\mathbf{Z}}_{\mathrm{gcn}}^{(j)}}{\tau}, \quad E_{ji}^{\mathrm{gcn} \to \mathrm{mp}} = \frac{\tilde{\mathbf{Z}}_{\mathrm{gcn}}^{(j)} \cdot \tilde{\mathbf{Z}}_{\mathrm{mp}}^{(i)}}{\tau}, \tag{11}$$

where $\tau$ is a temperature hyperparameter. A binary indicator matrix $Q \in \{0,1\}^{N \times N}$ specifies positive pairs, where $Q_{ij} = 1$ if and only if $i = j$.

We apply the InfoNCE loss in both directions to encourage a symmetric alignment between the two views:

**From semantic to structure:**

$$\mathcal{L}_{\text{mp}\rightarrow\text{gcn}} = -\frac{1}{N} \sum_{i=1}^{N} \log \frac{\exp(E_{ii}^{\text{mp}\rightarrow\text{gcn}})}{\sum_{j=1}^{N} \exp(E_{ij}^{\text{mp}\rightarrow\text{gcn}})}. \tag{12}$$

**From structure to semantic:**

$$\mathcal{L}_{\text{gcn}\rightarrow\text{mp}} = -\frac{1}{N} \sum_{j=1}^{N} \log \frac{\exp(E_{jj}^{\text{gcn}\rightarrow\text{mp}})}{\sum_{i=1}^{N} \exp(E_{ji}^{\text{gcn}\rightarrow\text{mp}})}. \tag{13}$$

The total contrastive loss is a weighted sum of the two directional losses, with a balancing weight $\lambda \in [0,1]$:

$$\mathcal{L}_{\text{contrast}} = \lambda \cdot \mathcal{L}_{\text{mp}\rightarrow\text{gcn}} + (1-\lambda) \cdot \mathcal{L}_{\text{gcn}\rightarrow\text{mp}}. \tag{14}$$

This bidirectional design ensures cross-view consistency and preserves the unique inductive biases of both the structural and semantic encoders, resulting in more robust and informative heterogeneous graph representations.

## 2.5 TIME COMPLEXITY ANALYSIS

The computational cost of DVCL comes from three components. (i) The schema view requires type-specific projection and R-GCN propagation, with complexity $\mathcal{O}\big(L_g(\sum_{a\in\mathcal{A}} |V^{(a)}|d_a + |E|)h\big)$, where $|V^{(a)}|$ is the number of nodes of type $a$, $d_a$ the feature dimension, $|E|$ the number of edges, $h$ the hidden size, and $L_g$ the number of R-GCN layers. (ii) The semantic meta-path Mamba module processes path-based sequences, yielding $\mathcal{O}(L_{\text{ml}}PL_ph)$, where $P$ is the number of meta-paths, $L_p$ the average path length, and $L_{\text{ml}}$ the number of stacked Mamba layers.(iii) The contrastive module performs projection and similarity computation, reduced by negative sampling to $\mathcal{O}(|V|h)$. In total, the framework has complexity $\mathcal{O}\big(L_g(\sum_{a\in\mathcal{A}} |V^{(a)}|d_a + |E|)h + L_{\text{ml}}PL_ph + |V|h\big)$, which scales linearly with the numbers of nodes, edges, and meta-path instances, ensuring efficiency on large heterogeneous graphs.

## 3 EXPERIMENTS

**Datasets.** we employ four public datasets, i.e., ACM Zhao et al. (2020) , DBLP Fu et al. (2020) , Freebase Li et al. (2021) , and Academic Hu et al. (2020a) to validate the performance of our method.

**Comparison methods.** To assess the effectiveness of our approach, we compare with representative baselines from three categories: (1) supervised methods, such as HGT Hu et al. (2020b); (2) generative methods, e.g., HGMAE Tian et al. (2023), DiffGraph Li et al. (2025); and (3) contrastive methods, including HeCo Wang et al. (2021), HDMI Jing et al. (2021), ASHGCL Jiang et al. (2025) .

**Settings.** All experiments were conducted on a single NVIDIA RTX 4090 GPU (24GB). Baselines were tuned via their official implementations for fair comparison. Reported results are averages over multiple runs.

Hyperparameters were tuned via grid search Bashir et al. (2016). For the GCN branch, we set the number of layers to $L_g = 3$. The Mamba branch employed $L_{\text{ml}} = 4$ stacked layers with a Hop2Token hop size of $K = 5$. The learning rate was fixed at $1 \times 10^{-3}$. The attention temperature $\tau$ and the contrastive loss weight $\lambda$ were selected from $\{0.1, 0.2, \ldots, 0.9\}$, while the hidden dimension was chosen from $\{64, 128, \ldots, 1024\}$. All models were trained for 100 epochs with early stopping based on validation performance. To ensure reproducibility, results were averaged over five

independent runs with random seeds drawn from $\{0, 42, 123, 2022, 3147\}$. For node classification, we report Macro-F1 (Ma-F1), Micro-F1 (Mi-F1), and AUC. For node clustering, performance is evaluated using normalized mutual information (NMI) and adjusted Rand index (ARI).

## 3.1 EXPERIMENTAL RESULTS AND ANALYSIS

**Classification** For the downstream classification task, we fine-tune the node representations obtained from our model and baselines using a logistic regression classifier, ensuring reproducibility by explicitly specifying the classifier choice. Following prior work, we randomly sample 20, 40, and 60 labeled nodes per class for training, while using 1,000 nodes each for validation and testing. The overall performance is summarized in Table 1.

DVCL achieves consistent gains across four benchmarks. On ACM (40 labels/class), it records 94.76% Ma-F1 and 94.91% Mi-F1, surpassing HGMS by over 2% thanks to effective semantic–structural alignment that enhances class discriminability. On the heterophilic Academic dataset, DVCL outperforms HERO by 12.57% (Ma-F1) and 7.37% (Mi-F1), demonstrating its ability to exploit informative cross-type relations while mitigating noisy neighbors. On DBLP and Freebase, DVCL secures top-2 performance, confirming that the combination of fine-grained semantic modeling and global structural alignment ensures robust generalization to both homophilous and heterophilic graphs.

| Dataset | Metric | Split | HGT WWW'20 | HeCo KDD'21 | HDMI WWW'21 | HGMAE AAAI'22 | HeCo++ TKDE'23 | RMR KDD'24 | HERO ICLR'24 | GTC NN'25 | ASHGCL IF'25 | DiffGraph WSDM'25 | HGMS SIGIR'25 | DVCL |
|---|---|---|---|---|---|---|---|---|---|---|---|---|---|---|
| ACM | Ma-F1 | 20 | 76.08±1.7 | 86.32±0.8 | 89.89±0.7 | 88.53±0.7 | 89.33±0.5 | 89.02±0.3 | 88.42±1.4 | 90.20±0.7 | 91.43±0.2 | 90.81±0.9 | **93.11±0.4** | 92.80±0.5 |
| | | 40 | 75.48±1.2 | 87.27±0.6 | 89.01±0.5 | 87.86±0.6 | 88.70±0.7 | 88.67±0.4 | 87.91±1.2 | 88.92±0.6 | 91.25±0.1 | 89.96±0.9 | 92.49±0.3 | **94.76±0.8** |
| | | 60 | 78.96±1.9 | 88.54±0.9 | 90.76±0.9 | 89.12±0.3 | 89.51±0.7 | 87.96±0.4 | 89.55±0.6 | 89.91±0.2 | 92.46±0.1 | 91.73±0.7 | 92.34±0.2 | **93.41±0.6** |
| | Mi-F1 | 20 | 77.13±1.6 | 89.32±0.8 | 89.89±0.4 | 89.21±0.7 | 88.96±0.5 | 88.94±0.3 | 88.67±0.9 | 90.64±0.2 | 90.75±0.6 | 90.42±1.0 | **92.74±0.3** | 92.60±0.4 |
| | | 40 | 76.98±1.3 | 87.73±0.4 | 89.01±0.5 | 88.79±0.3 | 88.40±0.8 | 88.56±0.4 | 89.01±0.6 | 88.55±0.3 | 90.92±0.1 | 90.95±0.7 | 92.30±0.4 | **94.91±0.7** |
| | | 60 | 79.77±2.9 | 89.13±0.7 | 89.61±0.9 | 88.36±0.7 | 89.30±0.7 | 87.76±0.4 | 89.13±0.5 | 89.45±0.4 | 92.25±0.1 | 90.73±0.6 | 92.26±0.3 | **93.60±0.6** |
| | AUC | 20 | 88.48±1.6 | 96.43±0.4 | 97.63±0.7 | 96.45±0.3 | 97.25±0.2 | 97.09±0.1 | 97.38±0.4 | 97.58±0.1 | 97.86±0.2 | 97.56±0.3 | **98.56±0.2** | 98.54±0.6 |
| | | 40 | 87.92±1.2 | 96.72±0.3 | 96.64±0.8 | 97.23±0.5 | 97.08±0.2 | 96.75±0.1 | 96.51±0.3 | 97.54±0.2 | 98.24±0.3 | 97.72±0.3 | 98.48±0.3 | **99.19±0.3** |
| | | 60 | 90.33±1.7 | 96.55±0.4 | 97.61±0.5 | 97.88±0.2 | 97.50±0.2 | 95.38±0.6 | 97.33±0.1 | 97.82±0.1 | 98.08±0.2 | 97.66±0.2 | 98.36±0.2 | **99.02±0.1** |
| DBLP | Ma-F1 | 20 | 88.38±1.7 | 91.32±0.1 | 91.09±0.7 | 91.65±0.6 | 91.40±0.2 | 91.32±0.4 | 91.42±0.6 | 93.12±0.3 | 93.05±0.2 | 92.01±0.2 | **93.21±0.3** | 92.71±0.2 |
| | | 40 | 88.07±1.2 | 90.27±0.2 | 91.16±0.4 | 90.89±0.3 | 90.56±0.2 | 91.23±0.3 | 91.32±0.6 | 91.62±0.3 | 92.05±0.1 | 91.90±0.3 | 92.00±0.3 | **92.27±0.3** |
| | | 60 | 88.42±1.1 | 91.27±0.1 | 91.31±0.4 | 92.17±0.2 | 91.01±0.3 | 91.60±0.3 | 92.01±0.3 | 92.95±0.2 | 92.46±0.1 | 92.13±0.3 | 93.04±0.2 | **93.16±0.3** |
| | Mi-F1 | 20 | 88.74±1.1 | 91.76±0.2 | 91.89±0.6 | 91.33±0.2 | 92.03±0.1 | 91.97±0.5 | 92.13±0.3 | 93.67±0.3 | 93.51±0.4 | 92.90±0.4 | **93.83±0.2** | 93.09±0.2 |
| | | 40 | 88.65±1.3 | 90.27±0.2 | 90.31±0.4 | 91.47±0.3 | 90.87±0.2 | 91.15±0.3 | 91.17±0.2 | 92.02±0.3 | 92.42±0.2 | 92.01±0.5 | 92.14±0.1 | **92.81±0.3** |
| | | 60 | 89.96±0.9 | 91.27±0.3 | 91.61±0.5 | 91.74±0.2 | 91.86±0.2 | 92.34±0.2 | 91.58±0.3 | 93.61±0.2 | 93.68±0.2 | 92.65±0.5 | 93.65±0.1 | **93.75±0.2** |
| | AUC | 20 | 97.73±0.7 | 98.32±0.2 | **99.89±0.8** | 98.45±0.2 | 98.39±0.1 | 98.28±0.1 | 98.41±0.1 | 98.96±0.1 | 98.81±0.2 | 98.92±0.5 | 98.77±0.1 | 98.71±0.2 |
| | | 40 | 97.36±0.9 | 98.27±0.2 | 98.03±0.2 | 98.79±0.2 | 98.17±0.1 | 98.21±0.1 | **98.92±0.1** | 98.46±0.1 | 98.42±0.3 | 98.42±0.1 | 98.58±0.1 | 98.61±0.1 |
| | | 60 | 97.54±0.6 | 99.07±0.1 | 98.43±0.3 | 98.89±0.1 | 98.62±0.1 | 98.56±0.1 | 98.64±0.1 | 98.89±0.1 | 98.95±0.2 | 99.15±0.3 | 99.12±0.1 | **99.17±0.2** |
| Freebase | Ma-F1 | 20 | 54.67±2.5 | 59.32±0.8 | 55.89±0.7 | 60.11±1.3 | 59.87±1.0 | 60.42±1.1 | 56.42±1.3 | 60.40±1.5 | 63.02±0.5 | 62.93±1.1 | 61.99±1.2 | **64.32±1.3** |
| | | 40 | 55.64±2.3 | 61.37±0.9 | 54.61±0.9 | 61.23±0.9 | 61.33±0.5 | 61.51±0.7 | 57.33±0.7 | 60.20±0.9 | 62.68±0.3 | **64.87±0.8** | 62.30±1.0 | 62.69±1.8 |
| | | 60 | 56.63±2.5 | 60.27±0.9 | 61.61±0.5 | 61.22±1.7 | 60.86±1.0 | 61.26±0.9 | 57.02±0.5 | 60.81±1.2 | 60.95±0.6 | 63.21±1.4 | 63.74±0.7 | **64.33±1.9** |
| | Mi-F1 | 20 | 60.67±2.9 | 62.32±0.8 | 58.89±1.7 | 61.96±0.5 | 62.29±1.7 | 61.27±1.0 | 59.93±1.6 | 64.58±1.7 | 67.01±0.3 | 64.55±1.9 | 65.96±1.4 | **67.10±1.9** |
| | | 40 | 64.98±2.7 | 64.17±0.5 | 62.79±1.5 | 64.33±1.6 | 64.27±0.5 | 63.79±1.2 | 63.79±1.8 | 64.90±1.6 | 66.22±0.5 | 65.07±1.2 | 65.68±1.2 | **67.50±2.1** |
| | | 60 | 63.74±1.7 | 63.28±1.3 | 60.61±0.8 | 63.76±1.8 | 64.15±0.9 | 64.18±1.4 | 62.99±1.8 | 65.86±1.3 | 64.80±0.3 | 66.32±1.9 | 67.65±1.0 | **68.70±1.2** |
| | AUC | 20 | 70.36±2.3 | 75.37±0.8 | 74.89±1.1 | 75.33±0.6 | 76.68±0.7 | 75.39±0.8 | 74.29±1.6 | 75.21±0.9 | 77.29±0.5 | 77.52±0.9 | 77.05±1.4 | **77.61±1.4** |
| | | 40 | 75.79±1.4 | 78.43±0.4 | 73.61±1.3 | 74.61±0.8 | 77.51±0.3 | 73.28±0.5 | 73.97±1.7 | 77.10±1.5 | 78.35±0.4 | **79.79±0.8** | 79.14±1.1 | 78.03±0.3 |
| | | 60 | 75.60±1.2 | 78.27±0.5 | 73.61±0.7 | 78.55±0.7 | 78.27±0.7 | 77.84±0.6 | 74.01±1.6 | 76.25±1.4 | 78.19±0.4 | 79.12±1.3 | 79.24±0.8 | **79.66±2.0** |
| Academic | Ma-F1 | 20 | 69.99±1.6 | 70.52±1.2 | 67.89±1.7 | 68.77±1.7 | 72.09±1.4 | 70.36±1.7 | 70.98±0.8 | 75.80±1.5 | 75.73±0.9 | 76.32±1.3 | 76.33±1.1 | **77.64±1.9** |
| | | 40 | 73.44±1.3 | 71.27±0.9 | 70.61±0.9 | 70.65±0.6 | 73.25±1.7 | 71.69±1.3 | 71.34±1.2 | 77.75±1.3 | 77.93±0.7 | 78.01±1.1 | 79.16±0.8 | **80.31±1.9** |
| | | 60 | 72.73±1.3 | 73.99±0.9 | 73.15±0.2 | 74.33±1.8 | 75.87±1.0 | 74.28±1.0 | 76.02±1.9 | 77.31±1.7 | 78.21±1.1 | 77.71±1.9 | 77.94±0.7 | **79.01±1.6** |
| | Mi-F1 | 20 | 71.87±1.9 | 72.32±1.8 | 71.53±1.7 | 70.89±1.7 | 74.01±1.9 | 72.78±1.7 | 72.89±1.2 | 77.03±2.1 | 78.12±1.3 | 77.97±1.9 | 78.31±0.9 | **79.9±1.8** |
| | | 40 | 72.27±1.6 | 73.97±1.5 | 73.59±1.5 | 72.77±1.5 | 76.50±1.5 | 75.58±1.4 | 74.41±1.8 | 79.62±1.4 | 78.12±1.7 | 79.33±0.9 | **81.05±0.7** | 79.9±1.8 |
| | | 60 | 71.45±1.4 | 73.27±0.9 | 74.59±1.1 | 73.43±0.9 | 77.15±2.2 | 74.49±1.5 | 75.17±1.8 | 81.28±2.0 | 78.21±1.9 | 80.22±1.7 | 81.58±0.8 | **83.20±1.7** |
| | AUC | 20 | 90.38±0.9 | 92.03±0.6 | 91.89±0.7 | 90.19±0.8 | 92.21±0.5 | 91.83±0.9 | 91.66±0.8 | 93.02±0.5 | 94.32±0.8 | 93.77±1.3 | 95.22±0.4 | **95.32±0.9** |
| | | 40 | 91.76±1.0 | 91.47±0.3 | 91.15±0.5 | 90.77±0.7 | 92.45±0.3 | 91.60±0.5 | 92.42±0.3 | 93.43±0.6 | 93.01±0.7 | 94.97±0.9 | **95.47±0.2** | 94.99±0.8 |
| | | 60 | 91.27±0.6 | 91.27±0.2 | 90.99±0.7 | 90.82±0.7 | 92.56±0.7 | 91.34±0.6 | 92.33±0.4 | 93.54±0.4 | 93.11±0.7 | 94.57±0.4 | **95.22±0.1** | 95.11±0.6 |

Table 1: Classification results of all methods on four datasets (Avg. ± Std.). Bold for "the best", and underline for "the second best".

**Clustering** We evaluate node representation quality via node clustering by applying K-means Hamerly & Elkan (2003) on the embeddings generated by different methods. Table 2 reports the results, showing consistent trends with the node classification task. DVCL outperforms all baselines on ACM, DBLP, and Freebase. On ACM, it raises NMI and ARI by 9.93% and 9.26% over the strongest baseline, owing to its semantic–structural alignment that sharpens community bound-

aries and reduces clustering ambiguity. Consistent gains on DBLP and Freebase further validate its robust clustering ability across heterogeneous graphs. Comparable gains are observed on DBLP and Freebase. These results demonstrate DVCL's superior ability to capture complex semantics and generalize across downstream tasks.

| Datasets | ACM | | DBLP | | Freebase | |
|---|---|---|---|---|---|---|
| Metrics | NMI | ARI | NMI | ARI | NMI | ARI |
| HeCo | 56.87 | 56.94 | 74.51 | 80.17 | 16.14 | 16.90 |
| HDMI | 57.91 | 54.75 | 75.79 | 80.01 | 16.53 | 17.47 |
| HGMAE | 64.57 | 67.85 | 75.78 | 81.12 | 17.18 | 17.94 |
| HeCo++ | 60.82 | 60.09 | 77.36 | 81.23 | 16.67 | 17.48 |
| RMR | 54.37 | 45.65 | 76.83 | 80.91 | 18.33 | 18.15 |
| HERO | 58.41 | 57.89 | 77.89 | 81.76 | 15.34 | 15.56 |
| GTC | 63.26 | 69.30 | 78.10 | 82.89 | 18.70 | 21.31 |
| HGMS | 71.97 | 74.18 | 81.54 | 82.78 | 22.28 | 23.14 |
| **DVCL** | **79.11** | **81.05** | **87.36** | **88.65** | **24.05** | **24.36** |

Table 2: Clustering results of all methods on three datasets. Bold for "the best", and underline for "the second best".

**Embedding Visualization** To provide a more intuitive comparison, we visualize the distribution of node representations learned by different models using t-SNE Van der Maaten & Hinton (2008). In Figure 3, node categories are distinguished by color. As shown, the baseline models tend to produce representations with relatively blurred boundaries between categories. In contrast, our DVCL method achieves the most distinct separation, with nodes of the same class forming compact and well-defined clusters.

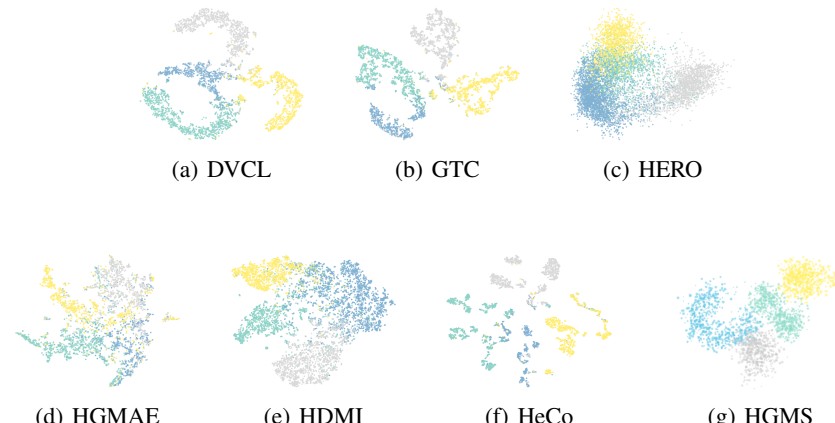

(a) DVCL      (b) GTC      (c) HERO

(d) HGMAE      (e) HDMI      (f) HeCo      (g) HGMS

Figure 3: Visualization of the embeddings learned by DVCL and baselines on DBLP dataset.

## 3.2 ABLATION STUDY

We conduct an ablation study to assess the role of each DVCL component (Figure 4). On both ACM and DBLP, the full DVCL achieves the best results, reaching 0.93/0.93/0.98 and 0.93/0.94/0.98 in Ma-F1/Mi-F1/AUC, respectively. Removing SMPMM (DVCL_NM) causes the largest drop (0.87/0.88/0.95 on ACM), underscoring its necessity for high-order semantic modeling. Replacing R-GCN (DVCL_NR) or removing bidirectional contrastive learning (DVCL_NB) also leads to consistent degradation, confirming the importance of relation modeling and contrastive regularization. These results demonstrate that each module is indispensable and jointly accounts for DVCL's superior performance.

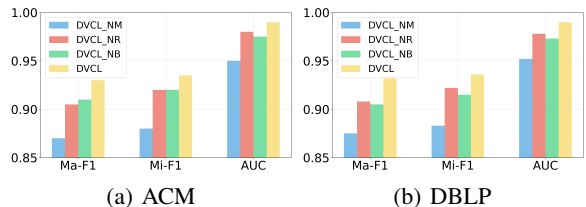

Figure 4: Classifications results of the proposed method with diverse variants.

### 3.3 HYPERPARAMETER ANALYSIS

We investigate three key hyperparameters, and the results are reported in Figure 5. As shown in Fig. 5(a) and Fig. 5(d), increasing $h$ from 64 to 128 enhances performance, but larger $h$ leads to degradation. This suggests that $h = 128$ already provides sufficiently expressive representations, while higher dimensions introduce redundancy and higher training costs, making $h = 128$ the best trade-off between effectiveness and efficiency. For $\tau$, which controls the weighting of contrastive paths, smaller values sharpen similarities and larger ones smooth them. Performance peaks at $\tau = 0.6$ (Fig. 5(b),(e)), showing it balances these effects most effectively. Finally, $\lambda$ tunes the balance between contrastive and supervised signals. A moderate value ($\lambda = 0.6$) consistently outperforms extremes, which bias learning toward a single signal, thus yielding stable and robust results across datasets.

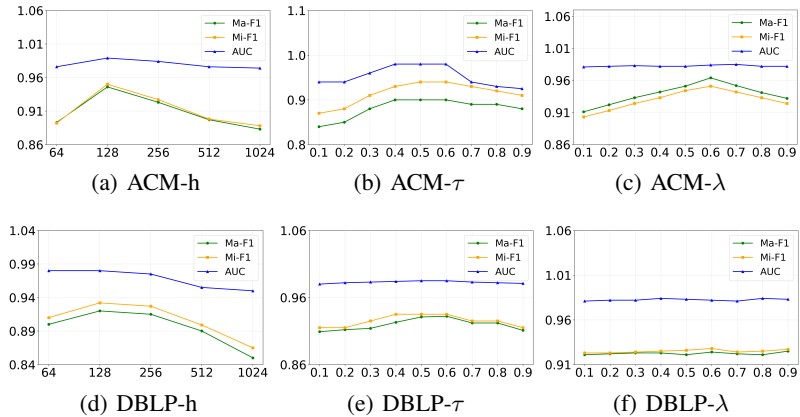

Figure 5: Classifications results of DVCL with different h, $\tau$ and $\lambda$ settings on ACM and DBLP.

## 4 CONCLUSION

In this paper, we propose DVCL, a modular and scalable framework for heterogeneous graph representation learning that unifies semantic and structural views via contrastive learning. DVCL couples a meta-path–aware Mamba module with a relational GCN schema module to capture long-range semantics and local structures, while a bidirectional contrastive objective aligns the two views without collapsing their inductive biases, producing robust and generalizable embeddings. Extensive experiments validate DVCL's superiority in node classification and clustering, with ablations underscoring the importance of SMPMM, relation-aware message passing, and contrastive fusion. Future work includes automated meta-path discovery and applications to heterogeneous brain networks and multi-modal neuroimaging to advance medical impact.

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

# A APPENDIX

## A.1 RELATED WORKS

Heterogeneous graph representation learning has received extensive attention, with existing works broadly categorized into supervised learning methods, generative approaches, and contrastive learning frameworks. While each line of research contributes toward modeling either structural patterns or semantic information, most still fall short of fully integrating both aspects in a unified and principled manner. Below, we review representative works from each category and discuss their connections to our proposed framework.

Supervised methods leverage annotated node and edge labels to guide representation learning. HGT Hu et al. (2020b) adopts type-specific attention mechanisms for scalable heterogeneous message passing, effectively capturing rich schema-level structures. HDMI Jing et al. (2021) enhances supervision by maximizing high-order mutual information between local and global contexts. Meta-path–based models such as HAN Wang et al. (2019) and MAGNN Fu et al. (2020) explicitly encode semantic dependencies through meta-path attention and aggregation. Despite their strong performance, these methods rely heavily on label availability and often emphasize structural propagation while underutilizing long-range semantic dependencies.

Generative approaches learn node representations by reconstructing features or relational structures without supervision. HGMAE Tian et al. (2023) reconstructs masked attributes and edge types to capture semantic dependencies. The generative-contrastive model in Wang et al. (2025b) integrates reconstruction with contrastive signals for improved semantic preservation. RMR Duan et al. (2024) proposes a selective masking mechanism that retains essential relational cues during reconstruction. Although effective, these approaches primarily focus on reconstruction-level objectives and may struggle to encode complex type-aware and high-order semantic dependencies inherent in heterogeneous graphs.

Contrastive methods have emerged as a powerful paradigm to address the scarcity of labels by maximizing agreement across multiple graph views. HeCo Wang et al. (2021) first introduced co-contrastive learning between schema-level and meta-path views. HeCo++ Liu et al. (2023) further incorporates hierarchical contrastive objectives across node types. GTC Sun et al. (2025) fuses GNNs and Transformers to jointly model local structures and global semantics, while HGMS Wang et al. (2025a) enhances homophily and smoothness through multi-view representation refinement.

Beyond these, several additional contrastive or multi-view frameworks further highlight the importance of capturing complementary information. DMGI Park et al. (2020) maximizes mutual information between local and global representations across multiple views, but it primarily targets homogeneous multiplex networks and lacks type-aware message passing. HGCML Wang et al. (2023a) employs multi-level contrastive objectives but derives all views from a shared encoder, potentially

---

[1]We acknowledge the use of a large language model (LLM) for assistance in polishing and refining the manuscript. However, the intellectual contribution and core significance of the work originate entirely from the authors.

entangling structural and semantic signals. PT-HGNN Jiang et al. (2021a) and CPT-HG Jiang et al. (2021b) adopt pre-training strategies to learn transferable heterogeneous graph representations, yet they rely on large-scale pre-training objectives and lack explicit cross-view semantic–structural alignment.

Despite these advancements, existing contrastive frameworks still suffer from three key limitations: (i) view construction and sampling strategies often depend on dataset-specific heuristics Yu et al. (2022); Hassani & Khasahmadi (2020); (ii) most methods struggle to capture high-order semantic dependencies along long meta-paths Wang et al. (2021); and (iii) structural and semantic information are often modeled through shared encoders or loosely coupled objectives, resulting in fragmented or modality-biased embeddings Jiao et al. (2020).

Unlike prior works that rely on global pre-training (e.g., PT-HGNN, CPT-HG) or generalized multi-view mutual information maximization (e.g., DMGI, HGCML), our proposed DVCL introduces a modular *dual-view contrastive* framework that: (i) models relation-aware structural dependencies via a schema-view R-GCN (GSVM), (ii) captures high-order semantic dependencies via a meta-path–aware Mamba module (SMPMM), and (iii) dynamically aligns them through bidirectional cross-view contrastive learning. This design enables principled, adaptive, and expressive integration of structural and semantic information, addressing the above limitations and achieving state-of-the-art performance across multiple benchmarks.

## A.2 EXPERIMENTS

### A.2.1 DATASETS

Four benchmark academic datasets are utilized to evaluate the clustering effectiveness of the proposed method. A summary of the datasets is presented in Table 3, and detailed descriptions of each dataset are as follows:

| Dataset | Node | Relations | Metapaths | Classes |
|---------|------|-----------|-----------|---------|
| ACM | paper (P): 4019
author (A): 7167
subject (S): 60 | P-A: 13407
P-S: 4019 | PAP
PSP | 3 |
| DBLP | author (A): 4057
paper (P): 14328
conference (C): 20
term (T): 7723 | P-A: 19645
P-C: 14328
P-T: 85810 | APA
APCPA
APTPA | 4 |
| Freebase | movie (M): 3492
actor (A): 33401
direct (D): 2502
writer (W): 4459 | M-A: 65341
M-D: 3762
M-W: 6414 | MAM
MDM
MWM | 3 |
| Academic | author (A): 28646
paper (P): 21044
venue (V): 18 | A-P: 69311
P-P: 21357
P-V: 21044 | APA
APPA
APVPA | 5 |

Table 3: The basic details of the datasets.

- **ACM** [2] comprises 3,025 research papers across 5 major computer science domains, with over 10,000 citation relationships. The dataset includes multi-modal features, such as textual representations derived from paper abstracts (2,000-dimensional), as well as citation-based interactions (binary relationships between paper pairs).

- **DBLP** [3] contains over 4,000 publications across a range of computer science subfields, including both conference and journal papers. The dataset is characterized by textual features (1,000-dimensional), co-authorship information (500-dimensional), and citation relationships among the papers.

- **Freebase** [4] is a large-scale knowledge base containing approximately 2.5 million entities and 18 million relationships. Features include entity attributes (2,500-dimensional),

---

[2] https://dl.acm.org/artifacts/dataset.

[3] https://dblp.uni-trier.de/xml/.

[4] https://developers.google.com/freebase.

category-based attributes (1,000-dimensional), and inter-entity relationships (1,000-dimensional).

- **Academic** [5] comprises over 10 million academic articles spanning a wide range of disciplines. The dataset features multi-modal information, including textual content (3,000-dimensional), citation relationships (binary), and co-author collaboration data (500-dimensional).

### A.3 COMPARISON METHODS.

To comprehensively assess the effectiveness of our proposed framework, we compare DVCL against a broad set of state-of-the-art baselines spanning three paradigms:

- **Supervised-based method:**
  - **HGT** Hu et al. (2020b): A supervised transformer-based model designed for heterogeneous graphs. It utilizes type-specific parameters and attention mechanisms to perform message passing across node and edge types under full label supervision.

- **Generative-based methods:**
  - **RMR** Duan et al. (2024): A generative framework that preserves relational semantics by reconstructing type-aware structure under multi-view settings. It captures structural signals via masked autoencoding.
  - **HGMAE** Tian et al. (2023): A heterogeneous graph masked autoencoder that leverages masked node and relation reconstruction to learn generalizable representations, focusing on intra- and inter-type dependencies.

- **Contrastive-based methods:**
  - **GTC** Sun et al. (2025): A graph topology contrastive framework that introduces topology-aware view generation and utilizes contrastive learning to enhance structural representation.
  - **HGMS** Wang et al. (2025a): A homophily-aware contrastive framework that enhances view quality via connection-strength–guided edge dropout and introduces a multi-view self-expressive module to infer homophilous relations and mitigate false negatives.
  - **HERO** Mo et al. (2024): A meta-path–free framework that captures both homophily and heterogeneity through self-expressive learning and heterogeneous message aggregation, with consistency and specificity losses to fuse complementary information.
  - **HeCo++** Liu et al. (2023): An extension of HeCo with hierarchical semantic modeling. It captures fine-grained dependencies across multiple semantic levels by constructing multi-resolution graph views.
  - **HeCo** Wang et al. (2021): A pioneering heterogeneous contrastive model that aligns structural and semantic views, typically constructed via meta-paths, to learn node embeddings without supervision.
  - **HDMI** Jing et al. (2021): A hybrid contrastive framework that bridges structure-level and instance-level contrast, jointly optimizing view consistency and semantic discrimination across modalities.

### A.3.1 SETTINGS.

We adopt the following widely used evaluation metrics to assess the performance of models on node classification and clustering tasks:

- **Macro-F1 (Ma-F1)** Opitz & Burst (2019): Measures the unweighted mean of F1 scores computed for each class individually. It is defined as:

$$\text{Ma-F1} = \frac{1}{C} \sum_{c=1}^{C} \text{F1}_c,$$

---

[5]https://www.microsoft.com/en-us/research/project/open-academic-graph/.

where $C$ is the number of classes, and $\text{F1}_c$ denotes the F1 score for class $c$. This metric treats all classes equally, making it suitable for imbalanced label distributions.

- **Micro-F1 (Mi-F1)** Opitz & Burst (2019): Computes the global F1 score by aggregating true positives (TP), false positives (FP), and false negatives (FN) across all classes:

$$\text{Mi-F1} = \frac{2 \cdot \sum_c \text{TP}_c}{2 \cdot \sum_c \text{TP}_c + \sum_c \text{FP}_c + \sum_c \text{FN}_c}.$$

It emphasizes performance on frequent classes and gives a better estimate of overall classification quality.

- **AUC (Area Under the ROC Curve)** Ma et al. (2024): Evaluates the trade-off between true positive rate (TPR) and false positive rate (FPR) across different classification thresholds. For binary classification, AUC is defined as:

$$\text{AUC} = \int_0^1 \text{TPR}(t) \, d\text{FPR}(t),$$

where $t$ is the decision threshold. AUC values closer to 1 indicate better class separability.

- **Normalized Mutual Information (NMI)** Estévez et al. (2009): Quantifies the similarity between clustering results $\mathcal{C}$ and ground-truth labels $\mathcal{Y}$, normalized by the geometric mean of their entropies:

$$\text{NMI}(\mathcal{C}, \mathcal{Y}) = \frac{2 \cdot I(\mathcal{C}; \mathcal{Y})}{H(\mathcal{C}) + H(\mathcal{Y})},$$

where $I(\cdot; \cdot)$ is mutual information and $H(\cdot)$ denotes entropy. NMI is invariant to label permutations and bounded in $[0, 1]$.

- **Adjusted Rand Index (ARI)** Santos & Embrechts (2009): Adjusts the Rand Index to account for chance groupings. Given a contingency table between clustering $\mathcal{C}$ and ground-truth $\mathcal{Y}$, ARI is computed as:

$$\text{ARI} = \frac{\sum_{ij} \binom{n_{ij}}{2} - \left[ \sum_i \binom{a_i}{2} \sum_j \binom{b_j}{2} \right] / \binom{n}{2}}{\frac{1}{2} \left[ \sum_i \binom{a_i}{2} + \sum_j \binom{b_j}{2} \right] - \left[ \sum_i \binom{a_i}{2} \sum_j \binom{b_j}{2} \right] / \binom{n}{2}},$$

where $n_{ij}$ is the number of samples in both cluster $i$ and class $j$, $a_i = \sum_j n_{ij}$, $b_j = \sum_i n_{ij}$, and $n$ is the total number of samples. ARI ranges from $-1$ to 1, with higher values indicating better clustering quality.

## A.4 ABLATION STUDY

In this section, we present an ablation study to rigorously assess the contribution of each core component in DVCL. We generate three distinct model variants by systematically removing or replacing essential modules, and compare their performance against the complete DVCL framework. The results of this comparison are visualized in Figure 6.

The three variants evaluated are as follows:

- **DVCL_NM**: This variant replaces the SMPMM with a standard Mamba module, focusing solely on global dependencies.

- **DVCL_NR**: In this variant, we substitute the R-GCN module with a conventional GCN, omitting the specialized modeling of heterogeneous relations.

- **DVCL_NB**: This variant replaces the bidirectional contrastive learning scheme with a unidirectional one.

As evidenced by the results, the full DVCL model consistently outperforms all ablated variants across multiple evaluation benchmarks, underscoring the synergistic benefits of our holistic design. Specifically, when the SMPMM module is replaced by Mamba (DVCL_NM), there is a significant performance drop, highlighting the importance of SMPMM in capturing task-specific relational dependencies—something that Mamba alone cannot address. Similarly, replacing R-GCN

with a standard GCN (DVCL_NR) leads to a noticeable degradation in performance, emphasizing the critical role of modeling heterogeneous relations for structure-aware representation learning. Finally, the substitution of the bidirectional contrastive learning mechanism with a unidirectional one (DVCL_NB) also results in substantial performance loss, validating the necessity of bidirectional alignment for enhanced feature representation.

In conclusion, the ablation study unequivocally demonstrates the indispensability of each component in DVCL. Each module—global sequence modeling through Mamba, structural relational modeling via R-GCN, and bidirectional contrastive learning—plays a vital role, and the removal or alteration of any module leads to considerable degradation in performance. This confirms the efficacy of our integrated approach, where these components work synergistically to optimize performance.

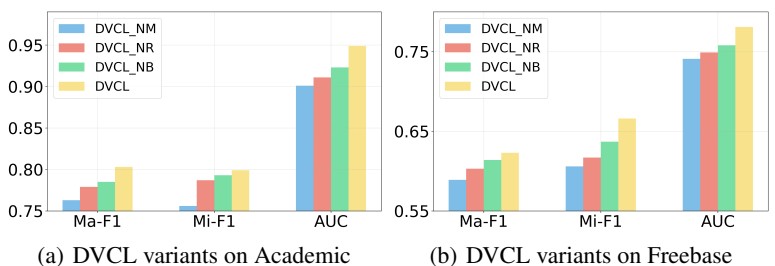

(a) DVCL variants on Academic    (b) DVCL variants on Freebase

Figure 6: Classifications results of the proposed method with diverse variants.

## A.5  HYPERPARAMETER ANALYSIS

The evaluation of the proposed method was conducted under varying hyperparameters across two benchmark datasets, Academic and Freebase. The results are presented in six subfigures, each corresponding to a different hyperparameter: node embedding dimension (h), temperature ($\tau$), regularization weight ($\lambda$), and the number of Mamba layers.

**(a) and (d) Academic-h, Freebase-h:** These subfigures assess the impact of the node embedding dimension. The results indicate that increasing the embedding dimension initially improves performance, but after a certain threshold, performance begins to degrade. Specifically, both the Ma-F1 and Mi-F1 scores peak at smaller dimensions (e.g., 128 or 256) and decline at larger dimensions (e.g., 1024). This observation suggests that excessively high dimensionality may lead to overfitting or result in unnecessary complexity that the model fails to effectively utilize, thus diminishing its performance. Hence, a moderate embedding dimension appears to be optimal for balancing representational power and generalization capability.

**(b) and (e) Academic-$\tau$, Freebase-$\tau$:** The temperature parameter ($\tau$) modulates the sharpness of the softmax distribution in the contrastive loss function. Across both datasets, performance remains relatively stable with only slight fluctuations as $\tau$ varies. This indicates that the model's performance is not highly sensitive to changes in $\tau$ within the considered range, demonstrating its robustness to this hyperparameter.

**(c) and (f) Academic-$\lambda$, Freebase-$\lambda$:** The regularization parameter ($\lambda$) regulates the trade-off between fitting the model to the data and avoiding overfitting. A higher $\lambda$ slightly reduces performance, especially in the AUC metric, across both datasets. This suggests that stronger regularization may restrict the model's ability to capture the data's complexity. However, the Ma-F1 and Mi-F1 scores remain relatively stable, implying that the model can still generalize effectively under moderate regularization.

**(g) and (h) Academic-ml, Freebase-ml:** These subfigures analyze the effect of the number of Mamba layers. Across both datasets, performance improves as the number of layers increases from 1 to 4, with a noticeable peak in the Ma-F1 and Mi-F1 scores. However, as the number of Mamba layers exceeds 4, performance starts to plateau and eventually decline. This behavior suggests that while additional Mamba layers help capture more complex dependencies, excessive layers can introduce noise or redundancy, diminishing the model's ability to generalize effectively.

The analysis of these hyperparameters reveals that the node embedding dimension is the most influential factor affecting model performance, followed by the number of Mamba layers. In contrast, the temperature and regularization weight have minimal impact, suggesting that the model exhibits robust performance across moderate variations in these hyperparameters.

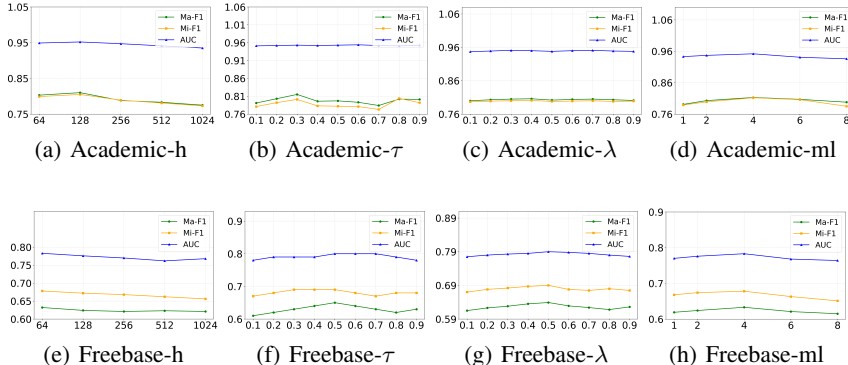

Figure 7: Classification results of DVCL with different h, $\tau$, $\lambda$ and mamba_layers settings on Academic and Freebase.

