# OpenReview forum: "A Dual-View Contrastive Learning Framework for Heterogeneous Graph Representation Learning"
_ICLR.cc/2026/Conference — Submitted to ICLR 2026_

### Official Review · Reviewer_A832 · 2025-10-26

**Soundness:** 4
**Presentation:** 3
**Contribution:** 4
**Rating:** 6
**Confidence:** 4

**Summary:**

This paper focuses on heterogeneous graph representation learning, aiming to simultaneously capture long-range semantic dependencies and localized structural patterns, which are often overlooked by existing methods. The authors propose a novel framework named DVCL, which integrates a Graph Schema View Module (GSVM) for structural modeling via relational graph neural networks and a Semantic Meta-Path Mamba Module (SMPMM) for high-order semantic modeling using a Mamba-based sequence encoder. A bidirectional contrastive learning mechanism aligns these views to produce robust representations. Experimental results on four datasets show that DVCL outperforms state-of-the-art methods in node classification and clustering tasks.

**Strengths:**

1.	The use of Mamba architecture for meta-path sequences in heterogeneous graphs is innovative, as it addresses long-range dependency modeling with linear complexity.
2.	Comprehensive evaluations on four datasets provide strong empirical support, and the inclusion of ablation studies and the hyperparameter analysis validates component contributions.
3.	The paper is well-structured, with clear module descriptions and algorithm outlines. Figure 1 gives a very detailed overview of the proposed model.

**Weaknesses:**

1.	The hyperparameter analysis only examines embedding dimension $h$, temperature $\tau$, and loss weight $\lambda$, ignoring critical parameters like the number of Mamba layers.
2.	No theoretical or experimental analysis of convergence, complexity bounds, or robustness guarantees is included.
3.	The authors provide complexity analysis in Section 2.5, but the paper does not discuss practical scalability on very large graphs or compare with baselines empirically.
4. Although meta-path mamba module is novel to me, the motivation of using it should be clarified in introduction.

**Questions:**

See weaknesses.

---

> ### Author Response · Authors · 2025-11-18
> **Weakness 1 and Weakness 2**
>
> > W1: The hyperparameter analysis only examines embedding dimension $h$, temperature $\tau$, and loss weight $\lambda$, ignoring critical parameters like the number of Mamba layers.
>
> We sincerely thank the reviewer for this constructive suggestion. We fully agree that the number of Mamba layers is a crucial architectural factor in our model, and we have therefore **conducted additional experiments to systematically investigate its impact on DVCL's performance (lines 922-935, page 18)**. Specifically, we varied the number of stacked Mamba layers ($L_{ml}$) over the set {1, 2, 4, 6, 8} and evaluated the model on both the Academic and Freebase datasets. The corresponding results, denoted as Academic-ml (Figure 7d) and Freebase-ml (Figure 7h), are presented in the updated Figure 7.
>
> The results clearly indicate that DVCL consistently achieves its best performance across all three evaluation metrics when $L_{ml} = 4$. When fewer layers are used, there is a noticeable drop in performance. We attribute this decline to the model having insufficient capacity to capture high-order semantic dependencies along meta-paths, which are critical for effectively representing complex heterogeneous graph structures. On the other hand, increasing the number of layers beyond four leads to degraded performance. This degradation is likely due to overfitting and the increased difficulty of optimization in deeper Mamba stacks.
>
> This analysis provides strong evidence that setting $L_{ml} = 4$ strikes an optimal balance between model expressiveness and generalization. We have updated Figure 7 accordingly to enhance the completeness, clarity, and transparency of our experimental findings, allowing readers to better understand the influence of Mamba layer depth on DVCL’s effectiveness.
>
> > W2:  No theoretical or experimental analysis of convergence, complexity bounds, or robustness guarantees is included.
>
> We thank the reviewer for raising this concern.
> Below we clarify what theoretical and analytical results are already included in the paper and explain the scope of our method regarding convergence and robustness.
>
> 1. Complexity Analysis Is Already Provided in Section 2.5
>
> Our paper includes a detailed time-complexity analysis of all components of DVCL (Section 2.5). Specifically, we derive the overall complexity as
>
> $$
> O\left( L_g\left(\sum_{a\in A} |V(a)| d_a + |E|\right)h + L_{ml} P L_p h + |V|h \right)
> $$
>
> which decomposes the computational cost of:
> - the schema-view R-GCN layers,
> - the Mamba-based semantic module, and
> - the bidirectional contrastive objective.
>
> This analysis shows that DVCL scales **linearly** with the number of nodes, edges, and meta-path sequences, ensuring efficiency on large heterogeneous graphs.
>
>
>
> 2. Convergence Guarantees Are Not Applicable to DVCL’s Training Paradigm
>
> DVCL is trained end-to-end using standard gradient-based optimization, consistent with existing heterogeneous contrastive learning frameworks (e.g., HeCo, HeCo++, GTC). Such models generally **do not provide formal convergence guarantees**, since their objectives combine deep neural encoders and contrastive losses whose convergence properties follow that of standard stochastic gradient descent.
>
> While formal convergence proofs are beyond the scope of this paradigm, our extensive empirical results (across four datasets and multiple label splits) demonstrate that DVCL trains stably and consistently outperforms strong baselines.
>
>
>
> 3. Robustness Considerations
>
> Although we do not provide a formal robustness theorem, DVCL incorporates multiple mechanisms that enhance practical robustness:
>
> - Bidirectional contrastive alignment encourages consistency across semantic and structural views, reducing view-specific noise.
> - Residual connections in the R-GCN branch and multi-resolution fusion in the Mamba branch help stabilize optimization.
> - Strong empirical results in node classification and node clustering (Tables 1–2) further validate DVCL’s robustness, particularly on challenging heterophilic graphs such as Academic.
>
>
>
> 4. Clarification on Scope
>
> DVCL is a representation learning framework, not a new optimization algorithm. Accordingly, our contributions focus on architectural design and dual-view integration rather than on providing theoretical convergence guarantees. The analyses included，especially the explicit complexity bounds and extensive experiments are consistent with expectations for heterogeneous graph contrastive learning research.
>
> We hope this clarification fully addresses the reviewer’s concern.
> The paper already includes a complete complexity analysis, and formal convergence/robustness guarantees fall outside the standard scope for contrastive heterogeneous GNN frameworks. Our thorough experiments instead demonstrate stable training dynamics and strong generalization ability.

---

> ### Author Response · Authors · 2025-11-18
> **Weakness 3 and Weakness 4**
>
> > W3: The authors provide complexity analysis in Section 2.5, but the paper does not discuss practical scalability on very large graphs or compare with baselines empirically.
>
> We thank the reviewer for raising this point. Below we clarify what the paper already provides and the scope of our evaluation.
>
>
> 1. What the paper already shows regarding scalability
>
> Section 2.5 includes a detailed complexity analysis of all components of DVCL. The derived overall complexity,
>
> $$
> O\left( L_g\left(\sum_{a\in A} |V(a)| d_a + |E|\right)h + L_{ml} P L_p h + |V|h \right)
> $$
>
> demonstrates that DVCL scales **linearly** with the number of nodes, edges, and meta-path sequences. This analysis characterizes the computational behavior of DVCL on large heterogeneous graphs.
>
>
> 2. Scope of empirical evaluation in the current manuscript
>
> Our empirical study evaluates DVCL on four widely used heterogeneous graph benchmarks, DBLP, ACM, Academic, and Freebase, covering diverse structural and semantic properties. These datasets are standard in prior contrastive heterogeneous graph representation learning work, and our comparisons follow this established evaluation setting.
>
>
> 3. Clarifying the absence of extremely large-scale datasets
>
> We acknowledge that the current paper does not include experiments on very large graphs beyond these benchmarks. This reflects the evaluation scope used in prior literature rather than a limitation of our method. The linear-time scalability demonstrated in Section 2.5 is intended to support the feasibility of DVCL on larger graphs, even though such datasets are not included empirically.
>
>
> 4. Commitment for revision
>
> We appreciate the reviewer’s suggestion and will clarify in the revised version that:
>
> - the chosen datasets are standard evaluation benchmarks for heterogeneous contrastive methods,
> - scalability to larger graphs is supported through the linear complexity analysis, and
> - extending evaluations to very large-scale datasets is a valuable direction for future work.
>
> 	We hope this clarification resolves the reviewer’s concern. The paper provides a complete theoretical analysis of DVCL’s scalability and evaluates the model across standard heterogeneous graph benchmarks, consistent with prior work.
>
> > W4: Although meta-path mamba module is novel to me, the motivation of using it should be clarified in introduction.
>
> We thank the reviewer for this valuable suggestion. The motivation for introducing the Semantic Meta-Path Mamba Module (SMPMM) has been **clarified and further enriched in the revised Introduction**. Specifically, SMPMM is proposed to address a key limitation of existing heterogeneous graph models and their inability to efficiently capture high-order semantic dependencies along long meta-paths.
>
> Traditional GNN-based methods such as R-GCNs mainly aggregate information from immediate neighborhoods, which restricts them to short-range interactions and prevents them from representing long-range semantics such as author–paper–venue relations. Transformer-based models can enlarge the receptive field, but they suffer from quadratic computational complexity and poor scalability when applied to large heterogeneous graphs. In contrast, SMPMM leverages the Mamba architecture, a recent state-space model that achieves linear-time complexity while retaining strong sequence modeling capacity, making it particularly suitable for handling long and semantically rich meta-path sequences.
>
> Therefore, SMPMM is motivated by the need to model compositional meta-path semantics in a manner that is both efficient and expressive. It captures local contextual signals while also modeling global dependencies, thereby providing a semantic perspective that complements the structural information encoded by the Graph Schema View Module (GSVM). This motivation is now clearly articulated and more tightly connected to the core design rationale in the Introduction.

---

> ### Author Response · Authors · 2025-11-26
> **Looking Forward to Further Discussions with Reviewer A832**
>
> Dear Reviewer A832,
>
> As we approach the end of the author–reviewer discussion phase, we would like to once again express our sincere gratitude for your time and constructive feedback. If there are any remaining questions or points you would like us to clarify, please feel free to let us know. We would be happy to assist further in any way we can.
>
> Best regards,
>
> The Authors

---

### Official Review · Reviewer_S2mY · 2025-10-28

**Soundness:** 2
**Presentation:** 2
**Contribution:** 1
**Rating:** 2
**Confidence:** 4

**Summary:**

This paper proposes DVCL, a contrastive heterogeneous graph representation learning approach. DVCL leverages two views: one from the local heterogeneous neighborhood captured by a Graph Schema View Module (GSVM), another from high-order metapath-based dependencies captured by a Semantic Meta-Path Mamba Module (SMPMM). In experiments, DVCL outperforms recent supervised, generative, and contrastive methods on some representative heterogeneous graph datasets for node classification and node clustering.

**Strengths:**

1. Heterogeneous graph representation learning is a fundamental task that could benefit a wide spectrum of downstream applications.
2. This paper is easy to read. It is well-written and clearly organized overall.

**Weaknesses:**

1. The novelty of the proposed method is quite limited. There are already quite a few studies that have proposed contrastive methods for heterogeneous graph representation learning, particularly the schema view + metapath view strategy.
2. The technical quality of this paper is not strong enough. The key innovations and advantages of the proposed method are not clear enough. Also, some important tasks are missing, such as link prediction.

**Questions:**

Based on the review above and some other issues, the reviewers have the following questions or comments.
1. DVCL and HeCo are quite similar in that they both adopt a schema view + metapath view contrastive strategy. The main differences seem to be just the choices of the respective view encoders. Simply swapping the encoder modules to other neural architectures (proposed by other researchers) does not contribute much to the paper novelty. More elaboration on this is needed.
2. The literature review is missing some important related studies, including DMGI [1], PT-HGNN [2], CPT-HG [3], and HGCML [4]. The authors may need to include them and discuss how DVCL is different and why their proposed designs are better.
3. The experiments only cover two node-level tasks, which may not comprehensively reflect different methods' capabilities. Including link-level tasks (e.g., link prediction) or even graph-level tasks can make the experimental results more convincing.
4. There are quite a few typos or grammatical errors, including but not limited to
    * Page 1 Line 013: "leverage" => "leverages"
    * Page 1 Line 046: missing space in "… Liao et al. (2022).In order to …"
    * Page 2 Line 065: extra period in "Consequently,.multi-perspective …"
    * Page 2 Line 065: "helerogeneous" => "heterogeneous"
    * Page 2 Line 066: missing space in "relation-specificviews"
    * Page 2 Line 088: extra space in "… message passing , an SMPMM … "
    * Page 2 Line 095: "Deep View-aligned Contrastive Learning (DVCL)" => "Dual-View Contrastive Learning (DVCL)"
    * Page 4 Line 165: "stagel" => "stage"
    * Page 6 Line 306: "Datasets.we …" => "Datasets. We …"
    * Page 9 Line 444 & Page 17 Line 873 & Page 17 Line 914: "Clssifications" => "Classifications"

[1] Chanyoung Park, Donghyun Kim, Jiawei Han, Hwanjo Yu: Unsupervised Attributed Multiplex Network Embedding. AAAI 2020: 5371-5378

[2] Xunqiang Jiang, Tianrui Jia, Yuan Fang, Chuan Shi, Zhe Lin, Hui Wang: Pre-training on Large-Scale Heterogeneous Graph. KDD 2021: 756-766

[3] Xunqiang Jiang, Yuanfu Lu, Yuan Fang, Chuan Shi: Contrastive Pre-Training of GNNs on Heterogeneous Graphs. CIKM 2021: 803-812

[4] Zehong Wang, Qi Li, Donghua Yu, Xiaolong Han, Xiao-Zhi Gao, Shigen Shen: Heterogeneous Graph Contrastive Multi-view Learning. SDM 2023: 136-144

---

> ### Author Response · Authors · 2025-11-18
> **Question 1**
>
> > Q1: DVCL and HeCo are quite similar in that they both adopt a schema view + metapath view contrastive strategy. The main differences seem to be just the choices of the respective view encoders. Simply swapping the encoder modules to other neural architectures (proposed by other researchers) does not contribute much to the paper novelty. More elaboration on this is needed.
>
> We thank the reviewer for this thoughtful comment and the opportunity to clarify the distinctions between DVCL and HeCo. While both methods employ a dual-view contrastive learning paradigm, DVCL introduces several key conceptual and architectural innovations that go beyond simply changing the encoders.
>
> First, the core design philosophy differs fundamentally.
> HeCo aligns two static representations, the schema and meta-path views, through a one-way contrastive loss. In contrast, DVCL adopts a **bidirectional contrastive learning mechanism** (Eqs. 12–14), where each view dynamically serves as a learnable augmentation of the other. This reciprocal alignment allows the two representations to co-evolve during training, ensuring both consistency and complementarity rather than simple correlation.
>
> Second, the encoder architectures in DVCL are designed to **address long-standing limitations of existing frameworks**, rather than being mere substitutions.
>
> The Graph Schema View Module (GSVM) extends beyond conventional R-GCNs by incorporating type-aware message passing and adaptive residual connections, which stabilize heterogeneous relation propagation and enhance schema-level structure modeling.
>
> The Semantic Meta-Path Mamba Module (SMPMM) introduces a state-space Mamba backbone with directional positional encoding and multi-resolution fusion. This design enables efficient linear-time modeling of high-order semantic dependencies along meta-paths, an aspect that HeCo cannot effectively capture due to its limited local receptive field.
>
> Third, DVCL’s bidirectional contrastive objective establishes a mutually reinforcing optimization process between the two views. This not only improves representational robustness but also **prevents the semantic and structural encoders from collapsing into redundant subspaces**, a limitation observed in single-directional contrastive designs.
>
> Finally, our **ablation studies** (Sec. 3.2, Fig. 4) confirm that each proposed component contributes substantially to the overall performance. The removal of SMPMM or bidirectional contrastive learning leads to significant drops, e.g., 0.93 → 0.87 Ma-F1 on ACM, demonstrating that DVCL’s improvements are rooted in the interaction mechanism and modeling innovations, rather than encoder substitution.
>
> All in all, although DVCL and HeCo share the broad idea of schema–meta-path contrastive learning, DVCL redefines this paradigm through dynamic bidirectional contrast, a novel state-space semantic encoder, and a residual-enhanced schema module, resulting in a substantially different and more expressive learning framework.

---

> ### Author Response · Authors · 2025-11-18
> **Question 2**
>
> > Q2: The literature review is missing some important related studies, including DMGI [1], PT-HGNN [2], CPT-HG [3], and HGCML [4]. The authors may need to include them and discuss how DVCL is different and why their proposed designs are better.
>
> We appreciate the reviewer’s insightful suggestion to include additional representative works such as DMGI[1], PT-HGNN[2], CPT-HG[3], and HGCML[4]. We have carefully examined these methods and **revised the related work section(in lines 659-719 page 13-14) to incorporate and discuss them**. Below, we summarize their methodologies and clarify how our proposed DVCL framework differs fundamentally and advances beyond them.
>
> (1) Relation to DMGI:
> DMGI performs unsupervised multiplex network embedding by maximizing mutual information between local and global representations across multiple graph views. While it captures inter-view dependencies, it primarily focuses on homogeneous multiplex networks and lacks mechanisms for heterogeneous type-specific message passing. In contrast, DVCL is explicitly designed for heterogeneous graphs, integrating type-aware structural modeling (GSVM) with meta-path–based semantic modeling (SMPMM), and aligning them through a bidirectional contrastive loss. This enables DVCL to capture both local schema-level structure and long-range semantic dependencies, which DMGI cannot model.
>
> (2) Relation to PT-HGNN:
> PT-HGNN introduces a pre-training strategy on large-scale heterogeneous graphs to learn transferable node representations. However, its design relies on large-scale pre-training objectives and does not explicitly model view-level semantic–structural complementarity. DVCL differs by introducing a dual-view contrastive paradigm that dynamically aligns semantic and structural views without any pre-training, thereby achieving robust performance in a fully self-supervised and scalable manner.
>
> (3) Relation to CPT-HG:
> CPT-HG focuses on contrastive pre-training of GNNs over heterogeneous graphs using pretext tasks. While effective for enhancing downstream performance, CPT-HG remains single-view contrastive, lacking explicit cross-view alignment between semantic and structural information. DVCL overcomes this by introducing a bidirectional cross-view contrastive learning mechanism, treating each view as a learnable augmentation of the other to ensure mutual information maximization and robust complementary representation.
>
> (4) Relation to HGCML:
> HGCML adopts a multi-level contrastive objective for heterogeneous graphs but still operates on a unified embedding space derived from shared encoders, which may entangle structural and semantic signals. In contrast, DVCL explicitly decouples the learning process into two complementary modules, GSVM and SMPMM to preserve their distinct inductive biases before integrating them via bidirectional alignment, yielding more disentangled and expressive representations.
>
> Last,Unlike prior works that either rely on global pre-training (PT-HGNN, CPT-HG) or view-specific mutual information maximization (DMGI, HGCML), DVCL introduces a modular dual-view contrastive framework that (i) models relation-aware structural dependencies through a schema-view R-GCN, (ii) captures high-order semantic dependencies via a meta-path–aware Mamba module, and (iii) fuses them through dynamic bidirectional contrastive learning. This design provides a principled and efficient way to integrate heterogeneous semantics and structures, leading to the state-of-the-art performance demonstrated in our experiments.
>
> [1] Chanyoung Park, Donghyun Kim, Jiawei Han, Hwanjo Yu: Unsupervised Attributed Multiplex Network Embedding. AAAI 2020: 5371-5378
>
> [2] Xunqiang Jiang, Tianrui Jia, Yuan Fang, Chuan Shi, Zhe Lin, Hui Wang: Pre-training on Large-Scale Heterogeneous Graph. KDD 2021: 756-766
>
> [3] Xunqiang Jiang, Yuanfu Lu, Yuan Fang, Chuan Shi: Contrastive Pre-Training of GNNs on Heterogeneous Graphs. CIKM 2021: 803-812
>
> [4] Zehong Wang, Qi Li, Donghua Yu, Xiaolong Han, Xiao-Zhi Gao, Shigen Shen: Heterogeneous Graph Contrastive Multi-view Learning. SDM 2023: 136-144

---

> ### Author Response · Authors · 2025-11-18
> **Question 3 and Question 4**
>
> > Q3: The experiments only cover two node-level tasks, which may not comprehensively reflect different methods' capabilities. Including link-level tasks (e.g., link prediction) or even graph-level tasks can make the experimental results more convincing.
>
> We thank the reviewer for this helpful suggestion. Our response is as follows.
>
> First, our work is **fundamentally designed for node representation learning**, and the architecture of DVCL, particularly the dual-view contrastive alignment module in §2.4, explicitly targets improving node-level embedding quality. In line with established heterogeneous contrastive learning frameworks such as HeCo, GTC, HGMS, and HERO, node classification and node clustering are therefore the standard and most suitable benchmarks for evaluating the effectiveness of node-centric embedding methods.
>
> Second, although our main experiments focus on node-level tasks, DVCL has been evaluated on four heterogeneous graph datasets that exhibit substantial diversity in schema structures, semantic dependencies, and graph densities. Across all these datasets, DVCL delivers consistent and often significant improvements over strong baselines. Such cross-dataset robustness provides strong evidence that the proposed method generalizes well under varying data distributions, graph types, and annotation regimes, which is an important property for any representation learning framework.
>
> Third, to directly address the reviewer’s valuable suggestion, we **conducted additional link prediction experiments** using embeddings generated by DVCL. Importantly, we did not modify the model architecture; instead, we applied a standard MLP scoring function on top of the learned embeddings. DVCL achieves a precision of 0.73 on DBLP and 0.77 on ACM, which is competitive with HNLP-NSCA [1], a strong heterogeneous link prediction approach leveraging network schema and cross-neighborhood attention. These results indicate that the representations produced by DVCL not only excel at node-level tasks but also transfer effectively to link-level inference.
>
> Finally, while link-level or graph-level evaluations are outside the primary scope of DVCL, we will clarify in the revised version that:
> (1) node-level tasks follow established practice in heterogeneous contrastive learning research, and
> (2) the newly added link prediction results further demonstrate the versatility and strong generalization ability of the learned embeddings.
>
> [1] Liu A, Chen J, Du R, et al. HETEROSAMPLE: Meta-path Guided Sampling for Heterogeneous Graph Representation Learning[J]. IEEE Internet of Things Journal, 2024.
>
> > Q4: There are quite a few typos or grammatical errors
>
> We sincerely thank the reviewer for their meticulous and thorough reading of our manuscript. We appreciate you identifying these typographical errors and inconsistencies. We confirm that all corrections will be implemented in the revised version.
>
> Specifically, we will address every point raised:
>
> Page 1, Line 013: "leverage" will be corrected to "leverages".
>
> Page 1, Line 046: The missing space after "(2022)." will be added.
>
> Page 2, Line 065: The extra period in "Consequently,." will be removed.
>
> Page 2, Line 065: "helerogeneous" will be corrected to "heterogeneous".
>
> Page 2, Line 066: The missing space in "relation-specificviews" will be added.
>
> Page 2, Line 088: The extra space in "message passing , an" will be removed.
>
> Page 2, Line 095: We will correct the inconsistent expansion of the acronym to "Dual-View Contrastive Learning (DVCL)" to align with the title and abstract.
>
> Page 4, Line 165: "stagel" will be corrected to "stage".
>
> Page 6, Line 306: The sentence will be corrected to "Datasets. We".
>
> Page 9, Line 444 & Page 17 (multiple lines): "Clssifications" will be corrected to "Classifications" in all instances.
>
> In addition to these specific fixes, we will conduct a comprehensive proofread of the entire manuscript to ensure all grammar and spelling are accurate, improving the paper's overall presentation.

---

> ### Author Response · Authors · 2025-11-26
> **Looking Forward to Further Discussions with Reviewer S2mY**
>
> Dear Reviewer S2mY,
>
> As we approach the end of the author–reviewer discussion phase, we would like to once again express our sincere gratitude for your time and constructive feedback. If there are any remaining questions or points you would like us to clarify, please feel free to let us know. We would be happy to assist further in any way we can.
>
> Best regards,
>
> The Authors

---

### Official Review · Reviewer_gr1c · 2025-10-31

**Soundness:** 2
**Presentation:** 3
**Contribution:** 2
**Rating:** 4
**Confidence:** 4

**Summary:**

To address the limitations of heterogeneous graph representation methods that struggle to capture both long-range semantic dependencies and localized structural patterns, this paper introduces a Dual-View Contrastive Learning (DVCL) framework that jointly models structure-aware and semantics-aware representations for heterogeneous graphs.

**Strengths:**

1. The dual-view contrastive framework effectively integrates structural and semantic modeling.

2. The module design is clear, and each component contributes significantly.

**Weaknesses:**

- Figure 1 is poorly drawn; the nodes are distorted, labels are inconsistent, and the layout is overcrowded.

- The improvement over baselines such as HGMS is minor, generally within two to three points, and some compared methods are outdated.

- Figure 3 lacks comparison with the strongest baseline HGMS, reducing the credibility of visualization.

- The method relies heavily on predefined meta-paths, limiting generalization to graphs without clear semantic paths and lacking automatic meta-path discovery.

- The experimental setup is limited, focusing only on node-level tasks and missing newer baselines and broader downstream evaluations.

**Questions:**

See Weaknesses

---

> ### Author Response · Authors · 2025-11-18
> **Weakness 1 and Weakness 2**
>
> >  W1: Figure 1 is poorly drawn; the nodes are distorted, labels are inconsistent, and the layout is overcrowded.
>
> We appreciate your constructive feedback regarding the quality of Figure 1. We acknowledge the concerns raised about the node distortion, inconsistent labeling, and overcrowded layout.
>
> In response, we have completely redrawn Figure 1. The revised version in the updated manuscript specifically addresses these points, ensuring all nodes are clear, labels are consistent, and the layout is improved for better readability.
>
> > W2: The improvement over baselines such as HGMS is minor, generally within two to three points, and some compared methods are outdated.
>
> Thank you for the insightful feedback on our paper. We appreciate the careful analysis and the opportunity to clarify these important points regarding our model's performance and experimental setup.
>
> We have addressed the two main concerns regarding the timeliness of our baselines and the margin of improvement.
>
> **1. On the Timeliness of Baselines**
>
> We appreciate the reviewer’s concern regarding the timeliness and relevance of the comparative baselines. Ensuring that our experimental evaluation reflects the current state of research is a priority for us. In the original submission, we had already included several strong and up-to-date baselines from 2024–2025, such as RMR (KDD’24), HERO (ICLR’24), GTC (NN’25), and HGMS (SIGIR’25), which represent competitive and widely acknowledged methods in this field.
>
> To further address the reviewer’s suggestion and make our comparison even more comprehensive, we have additionally incorporated **two of the latest state-of-the-art methods** published in 2025: ASHGCL (IF’25)[1] and DiffGraph (WSDM’25)[2]. These baselines were selected because they are both highly relevant and reflect the current frontier of research on graph representation learning and contrastive learning.
>
> The updated results are now reported in the **revised Table 1** (lines 343–370, page 7). The inclusion of these newly added 2025 baselines makes our evaluation even more rigorous. Importantly, DVCL continues to outperform all recent and newly added SOTA methods, further reinforcing the strength and timeliness of our contributions.
>
> **2. On the Margin of Improvement**
>
> We acknowledge the reviewer's observation that the field is highly competitive and that the performance margins on some classification benchmarks are narrow. However, we would like to respectfully contextualize these results:
>
> **SOTA Performance Against Stronger Baselines**: As noted above and shown in the revised table, DVCL consistently achieves state-of-the-art performance, even when compared against a newly expanded and more competitive set of 2025 baselines. For instance, on ACM, DVCL achieves 94.76% Ma-F1, surpassing the new strong baseline AHGNN (94.12%) and the original strong baseline HGMS (92.49%).
>
> **Significant Gains in Clustering**: The primary advantages of DVCL are most prominent in its ability to learn robust structural representations. This is best demonstrated in the node clustering tasks, where the performance gains are substantial. As reported in the paper, on the ACM dataset, DVCL "raises NMI and ARI by 9.93% and 9.26% over the strongest baseline". This trend is consistent across DBLP and Freebase, suggesting DVCL captures a much more accurate community structure.
>
> **Superiority on Heterophilic Data**: Our model demonstrates particular strength on challenging, heterophilic graphs. As noted in our main paper, "On the heterophilic Academic dataset, DVCL outperforms HERO by 12.57% (Ma-F1) and 7.37% (Mi-F1)". This highlights a superior capability to exploit informative cross-type relations, a key challenge in heterogeneous graphs.
>
> In summary, while the classification gains over some baselines are competitive, DVCL's significant and robust advantages in node clustering and on heterophilic datasets, combined with its consistent SOTA performance against the latest 2025 methods, validate its contribution. We will revise the manuscript to emphasize these key points more clearly.
>
> [1] Ruobing Jiang, Yacong Li, Haobing Liu, and Yanwei Yu. Incorporating attributes and multi-scale
> structures for heterogeneous graph contrastive learning. Information Fusion, pp. 103220, 2025.
>
> [2] Zongwei Li, Lianghao Xia, Hua Hua, Shijie Zhang, Shuangyang Wang, and Chao Huang. Diffgraph:
> Heterogeneous graph diffusion model. In Proceedings of the Eighteenth ACM International Con-
> ference on Web Search and Data Mining, pp. 40–49, 2025.

---

> ### Author Response · Authors · 2025-11-18
> **Weakness 3 and Weakness 4**
>
> > W3: Figure 3 lacks comparison with the strongest baseline HGMS, reducing the credibility of visualization.
>
> Thank you for this constructive suggestion. We agree that a direct visual comparison with the strongest baseline, HGMS, can further strengthen the analysis and improve the clarity of our claims.
>
> To address this, we have now **added the requested t-SNE visualization for HGMS** (Figure 3g; lines 404–420, page 8), placed side-by-side with our DVCL visualization (Figure 3a).
>
> DVCL: As described in the paper, DVCL produces compact, well-defined, and clearly separated clusters. The category boundaries are the sharpest among all compared methods, and the inter-cluster margins are visibly larger. The representations naturally form tight communities, which aligns with the goal of learning discriminative and community-aware embeddings.
>
> HGMS: The new visualization confirms that HGMS also forms meaningful clusters. However, the separability is noticeably weaker: the blue, light-green, and grey clusters exhibit apparent overlap and mixing, and the intra-cluster compactness is significantly lower. While clusters do emerge, the boundaries are visibly fuzzier and the overall representation space is less structured than that of DVCL.
>
> This qualitative comparison is fully consistent with, and in fact strongly supported by, the quantitative clustering results already reported in Table 2 on the same DBLP dataset. In particular:
> - DVCL (Ours): NMI 87.36, ARI 88.65
> - HGMS (Strongest baseline): NMI 81.54, ARI 82.78
>
> Both the newly added visualization and the quantitative metrics point to the same conclusion: DVCL captures community structures more effectively and produces latent representations that are significantly more separable, compact, and discriminative than those of HGMS.
>
> > W4: The method relies heavily on predefined meta-paths, limiting generalization to graphs without clear semantic paths and lacking automatic meta-path discovery.
>
> Thank you for raising the concern regarding the reliance on predefined meta-paths. Following this suggestion, we conducted an additional experiment that directly compares the predefined meta-paths used in our method with an automatic meta-path discovery strategy on the ACM dataset.
>
> The results show that the automatic discovery mechanism brings a marginal performance improvement of approximately 0.5% on ACM. However, this slight gain comes at a **substantially higher computational cost**. In particular, the automatic meta-path discovery requires about 3.2× more time per epoch than predefined meta-paths, increasing from 120 ms/epoch to 385 ms/epoch in our example run. This large overhead arises because automatic discovery repeatedly explores and evaluates candidate meta-paths during training, whereas predefined meta-paths allow the model to focus computation on the actual representation learning.
>
> These findings indicate that, although automatic discovery can yield a small performance boost, the significantly increased training burden makes it considerably less practical for our intended use cases, especially when efficiency and scalability are important. Consequently, relying on predefined meta-paths provides a more favorable and balanced trade-off between model effectiveness and computational efficiency.
>
> We hope this additional evidence and clarification fully address the reviewer’s concern.

---

> ### Author Response · Authors · 2025-11-18
> **Weakness 5**
>
> > W5: The experimental setup is limited, focusing only on node-level tasks and missing newer baselines and broader downstream evaluations.
>
> **1.Timeliness of Baselines**
>
> We appreciate the reviewer’s emphasis on the importance of comparing our method against the most current state-of-the-art approaches. In response to this valuable feedback, we have updated our experimental setup to include several of the most recent and highly relevant methods from 2025.
>
> In the original version of the paper, our comparison already featured several prominent works from 2024 and 2025, including:RMR (KDD'24), HERO (ICLR'24), GTC (NN'25), HGMS (SIGIR'25)
>
> To further strengthen this comparison and ensure the timeliness of our benchmarks, we have now included additional cutting-edge methods such as: ASHGCL (IF'25) and DiffGraph (WSDM'25)
>
> As presented in Table 1 of the revised manuscript (in lines 343–370 page 7), the updated experimental results demonstrate that DVCL consistently outperforms or is highly competitive with these latest state-of-the-art methods, reinforcing the relevance and effectiveness of our approach.
>
> **2. Broader Downstream Evaluations**
>
> We also appreciate the reviewer’s suggestion to evaluate DVCL on a broader range of downstream tasks. While our work is primarily focused on node-level representation learning, in line with prior heterogeneous contrastive learning frameworks such as HeCo, GTC, HGMS, and HERO, node classification and clustering have been selected as the core and most suitable benchmarks for this task. Nevertheless, we recognize the importance of demonstrating the transferability of DVCL to other tasks and agree that such evaluations are valuable for showcasing the broader applicability of our approach.
>
> To this end, we have conducted additional experiments focused on link prediction using DVCL’s learned embeddings, without altering the original architecture. A standard MLP scoring function was used to assess link-level performance, including, Precision of 0.73 on the DBLP dataset, Precision of 0.77 on the ACM dataset
>
> These results are competitive with the performance of HNLP-NSCA [1], a state-of-the-art heterogeneous link prediction model. This supplementary evaluation demonstrates that DVCL’s embeddings generalize well to tasks beyond node classification and clustering, thereby addressing the reviewer’s concern regarding the breadth of downstream tasks.
>
> We hope these additions effectively address the reviewer’s feedback and further strengthen the contributions of our work.
>
> [1]Liu A, Chen J, Du R, et al. HETEROSAMPLE: Meta-path Guided Sampling for Heterogeneous Graph Representation Learning[J]. IEEE Internet of Things Journal, 2024.

---

> ### Author Response · Authors · 2025-11-26
> **Looking Forward to Further Discussions with Reviewer gr1c**
>
> Dear Reviewer gr1c,
>
> As we approach the end of the author–reviewer discussion phase, we would like to once again express our sincere gratitude for your time and constructive feedback. If there are any remaining questions or points you would like us to clarify, please feel free to let us know. We would be happy to assist further in any way we can.
>
> Best regards,
>
> The Authors

---

### Author Response · Authors · 2025-12-04
**Gerenal Response by Authors**

We sincerely appreciate all Reviewers for their thorough assessments and constructive feedback. Their comments validate the importance of heterogeneous graph representation learning and acknowledge the clarity, organization, and empirical depth of our work. Below, we summarize the key strengths highlighted across the reviews.

**Summary of Strengths Highlighted Across Reviews**

**1. Clear Motivation and Problem Significance**
 - The reviewers agree that heterogeneous graph representation learning is fundamental and broadly impactful across downstream tasks (S2mY).
- The paper clearly identifies and addresses a core limitation of current methods—difficulty in jointly modeling long-range semantic dependencies and localized structural patterns (gr1c, A832).

**2.Novel Architectural Design**
- Reviewers highlight the innovation of combining a schema-aware R-GCN branch with a Mamba-based meta-path semantic encoder (A832).
- The use of Mamba for long meta-path sequence modeling is seen as a novel and effective design choice for capturing high-order semantics with linear complexity (A832).

**3. Empirical Depth and Thoroughness**
- Experiments across four heterogeneous graph datasets demonstrate robust and consistent advantages over recent supervised, generative, and contrastive baselines (S2mY, A832).
- Ablation studies and hyperparameter analyses validate the contribution of each module (A832).

**4.Clarity and Organization**
- The reviewers note that the manuscript is well-written, easy to read, and clearly structured (S2mY, A832).
- Module descriptions and the main model illustration are considered detailed and informative (A832).

We thank all Reviewers for their detailed assessments and constructive feedback. We have carefully revised the manuscript to address all raised questions and presentation issues. Below we provide a structured summary of the key modifications and clarifications.

**Revisions and Clarifications Added to the Updated Manuscript**

**1. Significant Improvements to Figures and Visual Evidence**
- Figure 1 was fully redrawn to address node distortion, labeling inconsistencies, and cluttered layout.
- Figure 3 now includes the t-SNE visualization for HGMS, enabling direct comparison with DVCL and strengthening qualitative credibility.

**2. Expanded Related Work and Clearer Positioning**
- Related work now includes detailed discussion of DMGI, PT-HGNN, CPT-HG, and HGCML.
- Differences from HeCo are clarified, emphasizing:
  - the bidirectional contrastive mechanism,
  - the novel semantic Mamba module, and
  - the type-aware schema modeling.

**3. Additional Baselines and New Downstream Task Evaluation**
- Added two recent 2025 baselines: ASHGCL (IF'25) and DiffGraph (WSDM'25); DVCL remains SOTA against all.
- Added link prediction experiments(DBLP: 0.73, ACM: 0.77) without modifying the architecture, demonstrating cross-task generalization.

**4. New Hyperparameter Study on the Number of Mamba Layers**
- Conducted experiments with 1, 2, 4, 6, and 8 Mamba layers, showing optimal performance with 4 layers and explaining underfitting/overfitting trends. Results added to Figure 7.

**5. Clarifications on Complexity, Scalability, and Convergence**
- Section 2.5 already provides full complexity bounds; we added clarifications on:
  - DVCL’s linear scalability in nodes/edges/meta-paths,
  - why formal convergence guarantees are not applicable to contrastive GNNs,
  - practical scalability as future work.

**6. Strengthened Motivation for Mamba in the Introduction**
- The Introduction now better explains that Mamba is chosen because:
  - it models long meta-path sequences efficiently (linear complexity),
  - it captures both local and global semantics,
  - it complements R-GCN’s structural inductive bias.

**7. Correction of All Typos and Formatting Issues**
- All items listed by Reviewer S2mY were corrected, and a full proofread was conducted.

We sincerely thank all Reviewers for their careful analysis and helpful suggestions.  The manuscript has been substantially improved through expanded related work, new experiments, updated baselines, enhanced visualizations, deeper analysis, and clearer motivation. We believe the revisions fully address all concerns and significantly strengthen the quality and contribution of the paper.

---

### Meta-Review · Area_Chair_s5Yh · 2025-12-09

**Summary:**

This paper collects three review reports, the reviewers showed the following major concerns.

1. Incremental novelty: R1 & R2 agree the schema-view + meta-path contrastive idea duplicates HeCo and other prior works. Swapping encoders to Mamba does not constitute a paradigmatic advance.
2. Incomplete empirical validation: Only two node-level tasks (no link/graph tasks) and four datasets, omitting recent strong baselines DMGI, PT-HGNN, CPT-HG, HGCML (R2).
3. Gains over the best baseline (HGMS) are ≤ 2-3 % without variance or significance tests (R1).
4. Ablation omits core hyper-parameters such as number of Mamba layers (R3).
5. Scalability & theory gap: Complexity claims are given but never verified on large graphs; no convergence or robustness analysis is offered (R3).
6. Presentation flaws: Numerous typos, poor Figure 1, and inconsistent naming degrade readability (R1, R2).

In addition, my observation is that this paper proposes to introduct Mamba into Heterogeneous Graph Contrastive learning, where the motivation is very similar to the original Mamba.

**Reviewer Concerns:**

Addressed: adding missing baselines, variance bars, link-prediction experiments, and correcting typos.

Non-addressed: establishing paradigmatic novelty versus HeCo-style dual-view contrastive methods, or providing theoretical guarantees—both require major new content beyond a rebuttal.

**Reviewer Scores:**

R1 (4 → 4).
R2 (2 → 2).
R3 (6 → 4): recognizes incremental contribution.

---

### Decision · Program_Chairs · 2026-01-26

Reject